# Multiplexed mRNA assembly into ribonucleoprotein particles plays an operon-like role in the control of yeast cell physiology

Rohini R Nair[1†], Dmitry Zabezhinsky[1†], Rita Gelin-Licht[1], Brian J Haas[2], Michael CA Dyhr[1], Hannah S Sperber[1], Chad Nusbaum[2‡], Jeffrey E Gerst[1*]

[1]Department of Molecular Genetics, Weizmann Institute of Science, Rehovot, Israel; [2]Broad Institute of MIT and Harvard, Cambridge, United States

**Abstract** Prokaryotes utilize polycistronic messages (operons) to co-translate proteins involved in the same biological processes. Whether eukaryotes achieve similar regulation by selectively assembling and translating monocistronic messages derived from different chromosomes is unknown. We employed transcript-specific RNA pulldowns and RNA-seq/RT-PCR to identify yeast mRNAs that co-precipitate as ribonucleoprotein (RNP) complexes. Consistent with the hypothesis of eukaryotic RNA operons, mRNAs encoding components of the mating pathway, heat shock proteins, and mitochondrial outer membrane proteins multiplex in trans, forming discrete messenger ribonucleoprotein (mRNP) complexes (called *transperons*). Chromatin capture and allele tagging experiments reveal that genes encoding multiplexed mRNAs physically interact; thus, RNA assembly may result from co-regulated gene expression. Transperon assembly and function depends upon histone H4, and its depletion leads to defects in RNA multiplexing, decreased pheromone responsiveness and mating, and increased heat shock sensitivity. We propose that intergenic associations and non-canonical histone H4 functions contribute to transperon formation in eukaryotic cells and regulate cell physiology.

**\*For correspondence:**
jeffrey.gerst@weizmann.ac.il

[†]These authors contributed equally to this work

**Present address:** [‡]Cellarity Inc, Cambridge, United States

## Introduction

Prokaryotic organisms can rely on polycistronic transcription (operons), which allows for the expression of needed mRNAs from a single promoter and enables a rapid and robust response to corresponding stimuli (e.g., lactose operon) (*Jacob and Monod, 1961*). However, this mode of action has not been reported in eukaryotes, except for a few limited examples (e.g., in *Caenorhabditis elegans* [*C. elegans*]) (*Spieth et al., 1993*), and eukaryotic operons have been observed to give rise to dicistronic messages (*Blumenthal, 2004*). However, it is intriguing to postulate whether eukaryotes have devised functional alternatives to operons. One possibility is that eukaryotic messenger ribonucleoprotein (mRNP) particles, which are composed of multiple RNAs and RNA-binding proteins (RBPs) (*Mitchell and Parker, 2014*), effectively confer combinatorial gene expression networks similar to prokaryotic operons (*Keene and Tenenbaum, 2002*). In this case, however, the mRNP particles contain individual mRNAs that undergo co-translational control and may encode proteins involved in the same biological process or molecular complex. The idea that mRNPs might essentially constitute RNA operons (*Keene, 2007*) or *transperons*, as termed here, implies several features. First, although mRNAs derived from different chromosomes can reside in the same mRNP, they should have common regulatory elements not only for transcriptional and translational control, but also for interacting at the post-transcriptional level with other mRNAs within the particle. The latter includes motifs/structures to facilitate interactions with shared RBPs, as well as elements that

might facilitate RNA-RNA interactions (e.g., base-pairing). Second, some mechanism must confer the recruitment and assembly, in trans (i.e., multiplexing), of the individual mRNAs into single mRNP particles. Third, in order to be functionally relevant, transperons should contain mRNAs encoding proteins involved in the same functional context (e.g., organelle biogenesis, macromolecular complex, or biological process). Although the mechanism remains unknown, studies have shown that genomic DNA folds create sites of transcriptional hot spots (*Lieberman-Aiden et al., 2009*; *Rao et al., 2014*) that could facilitate multiplexing by co-localizing messages during transcription and coordinating the subsequent association of RBPs. Moreover, RNA-RNA interactome studies show that extensive interactions can occur directly between RNAs in living cells (*Aw et al., 2016*; *Engreitz et al., 2014*; *Kudla et al., 2011*; *Nguyen et al., 2016*; *Sharma et al., 2016*). Thus, the multiplexing of mRNAs into mRNPs to yield transperons should be directly testable.

Although individual yeast RBPs have been shown to bind to numerous (e.g., 10s-1000s) mRNAs (*Hogan et al., 2008*), these RNA-binding studies are based primarily upon crosslinking and RBP pulldowns that are unable to define the minimal number of mRNA species in a single mRNP particle. To test the idea of RNA multiplexing and define the composition of such particles, we employed a single mRNA species pulldown procedure (RaPID) (*Slobodin and Gerst, 2010*; *Slobodin and Gerst, 2011*). This method employs MS2 aptamer tagging of endogenously expressed messages in yeast (*Haim et al., 2007*) and their precipitation from cell lysates via the MS2 coat protein (MCP) to identify bound non-tagged transcripts using RNA-seq (RaPID-seq). We have previously demonstrated that RaPID-seq identifies MS2-tagged target mRNAs (*Haimovich et al., 2016*) and now show here that additional messages associate with these transcripts.

By employing *MATα* yeast in RaPID-seq pulldown experiments, we found that select MS2-labeled target mRNAs (e.g., *SRO7*, *EXO70*, *OM45*) co-precipitated untagged mRNAs encoding secreted proteins involved in cell mating (e.g., *STE3*, *SAG1*, *MFα1*, *MFα2*). Since a corresponding complex (e. g., *STE2*, *AGA1*, *AGA2*, *MFA1*, *MFA2*) could also be precipitated from *MATa* cells, it suggested to us that mRNAs encoding secreted proteins involved in mating (e.g., pheromones, pheromone receptors, agglutinins, and proteases) multiplex into a functionally selective mRNP complex – the mating transperon. To identify potential factors involved in mating transperon assembly, we employed pulldowns of these mRNAs followed by mass spectrometry (RaPID-MS). We found that the yeast histone H4 paralogs, Hhf1 and Hhf2 (*Dollard et al., 1994*), interact with these mRNAs to regulate complex assembly. Furthermore, both pheromone responsiveness (e.g., shmooing) and mating were inhibited by either histone H4 depletion or a block in H4 acetylation. When mutated, no other histones had this effect. Moreover, we could identify conserved *cis* elements in the mating mRNAs and demonstrate that mutations therein had deleterious effects on mRNA multiplexing, pheromone responsiveness, and mating, which were similar to those seen upon histone H4 mutation.

To help elucidate the mechanism by which mRNA multiplexing might occur, we performed chromatin conformation capture (*Lieberman-Aiden et al., 2009*), as well as genomic locus tagging experiments, to look for evidence of direct allelic interactions. Importantly, a specific interaction between the *STE2* and *AGA2* genes was identified in *MATa* cells, which suggests that interallelic coupling may give rise to RNA multiplexing. Furthermore, coupling appeared to occur in both a transcription- and histone H4-dependent manner. To help validate this hypothesis, we examined whether mRNAs encoding heat shock proteins (HSPs), whose genes were previously shown to undergo intergenic interactions during heat shock (*Chowdhary et al., 2019*), also undergo multiplexing. Importantly, HSP mRNAs were found to multiplex in trans during heat shock to form RNP complexes, and histone H4 was required for both RNA multiplexing and robust post-heat shock cell recovery. Parallel studies revealed the existence of additional RNA multiplexes for mitochondrial outer membrane proteins and MAP kinase pathway proteins in yeast. Together, these results suggest that histone H4-mediated chromatin interactions may lead to the formation of functionally selective transperons in eukaryotic cells and possibly act as a means to co-regulate gene expression in place of polycistronicity.

## Results

### mRNAs encoding yeast mating pathway components co-precipitate in mRNPs

We developed the RaPID procedure (*Slobodin and Gerst, 2010*; *Slobodin and Gerst, 2011*) to identify RBPs and mRNAs that bind to specific RNAs of interest. Unlike CLIP or PAR-CLIP, which collects information on all RBP-RNA interactions in the cell and is biased toward highly expressed mRNAs with long poly-A tails (*Hafner et al., 2010*), RaPID allows for a biochemical view of the protein and RNA constituents of mRNPs at the single transcript level. RaPID employs the precipitation of MS2 aptamer-tagged messages using the MCP fused to streptavidin-binding peptide, followed by elution from immobilized streptavidin with free biotin, and then mass spectrometry (*Slobodin and Gerst, 2010*; *Slobodin and Gerst, 2011*) or RNA-seq (*Haimovich et al., 2016*) to identify bound proteins and RNAs, respectively. In order to address issues regarding whether the MS2 system might affect the stability of MS2 aptamer-tagged messages, we performed RaPID and RNA-seq (RaPID-Seq) on 11 endogenously expressed MS2 aptamer-tagged messages in *MAT*α yeast (*Haimovich et al., 2016*). These included representative transcripts having a wide range of intracellular patterns of localization (e.g., mitochondria: *OXA1* and *OM45*; cortical endoplasmic reticulum [cER]/asymmetrically localized mRNAs: *ABP1*, *SRO7*, *EXO70*, *ASH1*, *MYO2*, and *MYO4*; peripheral nuclear endoplasmic reticulum [nER]: *ATG8*; and peroxisomes: *PEX14*) (*Gadir et al., 2011*; *Haim et al., 2007*; *Zipor et al., 2009*). In addition to identifying the target mRNA in each pulldown, we identified non-tagged RNAs that co-precipitated with the targets (*Figure 1A*, *Figure 1—figure supplement 1*, and see *Supplementary file 1* for RNA-seq data). These included retrotransposable elements (e.g., Ty elements), tRNAs, ribosomal RNAs, telomere and small nuclear RNAs (*Figure 1—figure supplement 1*), and coding RNAs (*Figure 1*, *Figure 1—figure supplement 1*, and *Supplementary file 1*). Notably, five mRNAs encoding secreted and membrane proteins (mSMPs) of the mating process were found associated with a subset of target RNAs. These target transcripts consisted of two cER-localized mRNAs that encode polarity factors (mPOLs; *SRO7* and *EXO70*) (*Aronov et al., 2007*) and *OM45,* an mRNA that encodes a mitochondrial outer membrane protein (MOMP), which we have shown to localize to both mitochondria and ER (*Gadir et al., 2011*). Co-precipitated non-tagged mRNAs included *STE3*, which encodes the a-pheromone receptors, *MFα1* and *MFα2*, which both encode α-mating factor, *SAG1,* which encodes α-agglutinin, and *AFB1*, which encodes the a-pheromone blocker (*Figure 1B*). This result is consistent with the idea that specific RNAs can undergo multiplexing.

Importantly, the corresponding *MAT*a mating-type RNAs (e.g., *STE2*, *MFA1*, *MFA2*, *AGA2*, and *BAR1*), as well as other *MAT*a RNAs involved in mating (e.g., *STE6*, *ASG7*), were negatively enriched (depleted) in these pulldowns (*Figure 1A and B*). While this result was predicted, due to their lack of expression in *MAT*α cells, nonetheless, it validates the specific pulldown of the *MAT*α mating mRNAs.

We verified the RaPID-Seq results using RaPID followed by qRT-PCR (RaPID-qPCR) and found that the pulldown of *SAG1* mRNA co-precipitated cohort mRNAs of the mating pathway (e.g., *MFα1*, *MFα2*), but not unrelated mRNAs (e.g., *UBC6*, *WSC2*) (*Figure 2A*). We noted that the original target mRNAs (*SRO7*, *OM45*) were present in this pulldown, though *OM45* was lowly represented in the qRT-PCR data as those mRNAs directly related to mating. We note that *IST2* mRNA was significantly enriched in the pulldown from *MAT*α cells (*Figure 2A*), but not in pulldowns from *MAT*a cells (*Figure 2B*).

By employing RaPID-qPCR, we examined whether a corresponding set of mating pathway mRNAs (i.e., *STE2* [α-factor receptor], *AGA1* and *AGA2* [agglutinins], and *MFA1* and *MFA2* [a-factor]) could co-precipitate in *MAT*a cells. Indeed, the precipitation of *AGA1* mRNA using RaPID led to co-precipitation of the cohort mating pathway mRNAs (e.g., *STE2*, *AGA2*, *MFA1*, *MFA2*; *Figure 2B*), along with *SRO7*. Thus, we demonstrated that mRNAs encoding secreted proteins involved in mating undergo multiplexing in either yeast haplotype.

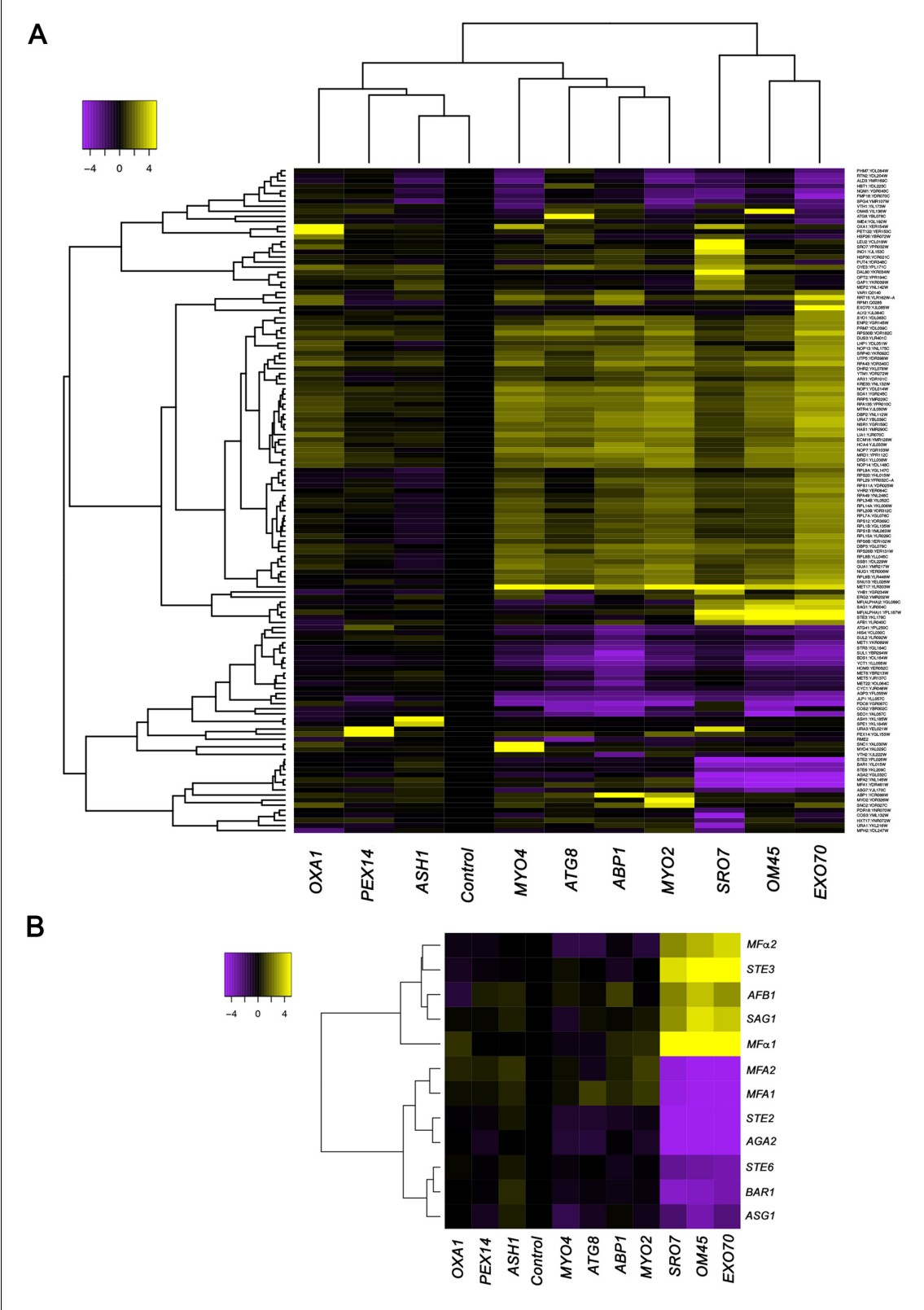

**Figure 1.** mRNAs encoding secreted and membrane proteins involved in yeast mating undergo multiplexing. (**A**) RaPID RNA pulldown and RNA-seq reveal transcript multiplexing in yeast. Different MS2 aptamer-tagged mRNAs (listed on the *x* axis) expressed from their genomic loci were precipitated from *MATα* yeast (BY4742) by MS2-CP-GFP-SBP after formaldehyde crosslinking in vivo and cell lysis procedures (RaPID; see 'Materials and methods'). RNA-seq was performed and the reads were plotted to yield a heat map for the relative enrichment of the tagged as well as non-tagged mRNAs (see

*Figure 1 continued on next page*

*Figure 1 continued*

color key for approximate values). All non-coding RNAs (e.g., Ty elements, tRNAs, snoRNAs, ribosomal RNAs) were removed but are included in the heat map of *Figure 1—figure supplement 1*. A list of the log2 transformed and control-subtracted expression values, as plotted here, is given in *Supplementary file 1*. (B) mRNAs encoding proteins involved in mating in *MATα* cells are enriched in the pulldowns of *SRO7, EXO70,* and *OM45* mRNAs. A selective heat map comprising the mRNAs encoding secreted and membrane proteins involved in yeast cell mating is shown. Positive enrichment of cell type-specific *MATα* mRNAs (*STE3, MFα1, MFα2, SAG1, AFB1*) is shown in purple, while negative enrichment of the non-expressed *MAT*a mRNAs (*MFA1, MFA2, STE2, AGA2, BAR1, STE6, ASG7*) is shown in yellow.

The online version of this article includes the following figure supplement(s) for figure 1:

**Figure supplement 1.** mRNAs encoding secreted and membrane proteins involved in yeast mating multiplex into a single ribonucleoprotein particle.

## mRNAs encoding mating pathway components co-localize to ER in both vegetative-growing and pheromone-treated cells

It was important to determine whether mRNAs of the mating pathway localize to the ER in the cell body, like other mSMPs (*Kraut-Cohen et al., 2013*), or perhaps to the bud/shmoo tip, like asymmetrically localized mRNAs that encode polarity and secretion factors (*Aronov et al., 2007*; *Gelin-Licht et al., 2012*). A previous study has suggested that aptamer-tagged mating pathway mRNAs, like *MFA2*, localize to P-bodies located in (or near) the shmoo tip and that these granules are functional sites for transmitting the mating signal (*Aronov et al., 2015*). However, a more recent work by us (*Haimovich et al., 2016*) has shown that endogenously expressed MS2 aptamer-tagged *MFA2* mRNA does not localize to P-bodies and that P-bodies only form under conditions of mRNA overexpression, as used in *Aronov et al., 2015*. We examined the localization of endogenously expressed MS2-tagged *STE2* and *AGA1* mRNAs, along an ER marker, either mCherry-Scs2 (*Figure 2C*) or -Sec63 (*Figure 2D*), in both non-treated and α-factor (1 µM)-treated cells. We found that both mRNAs co-localized either with nER or with cER, both with or without α-factor (images for mCherry-Scs2 are shown in *Figure 2C* and quantified for mCherry-Sec63 in *Figure 2D*), and are not present in the bud or shmoo tips (*Figure 2C*). In addition, we examined the localization of endogenously expressed MS2 aptamer-tagged *MFA1* and *MFA2* mRNA in wild-type (WT) cells or cells lacking *SHE2*, an ER-localized RBP involved in the asymmetric localization of mRNAs to the bud (but not shmoo) tip (*Genz et al., 2013*). Both *MFA1* and *MFA2* mRNAs localized to the cell bodies in either pheromone-treated and non-treated cells, and no difference in localization was observed upon the deletion of *SHE2* (*Figure 2—figure supplement 1A*). Thus, like other mSMPs, these mating pathway mRNAs are not trafficked to the polarized extensions of yeast cells (*Kraut-Cohen et al., 2013*).

As RNA co-localization is predicted by RNA multiplexing, we tagged endogenous *AGA2* with the PP7 aptamer (*Larson et al., 2011*) in the cells expressing MS2-tagged *STE2* and examined for co-localization upon expression of their respective fluorescent protein-tagged aptamer-binding proteins (e.g., PP7-PS-tomato and MS2-CP-GFP(x3)). Overall, we found that 65.5 ± 7.7% (average ± standard deviation; n = 3 experiments) of PP7-tagged *AGA2* mRNA co-localized with MS2-tagged *STE2* mRNA (*Figure 2D*), of which 39.5 ± 3.5% co-localized in the cytoplasm and 60.5 ± 3.5% co-localized in the nucleus. We validated the live imaging results using single-molecule fluorescence in situ hybridization (smFISH), which is typically more sensitive than imaging using MS2-CP-GFP(x3) and thus reveals a greater number of mRNAs (*Figure 2E and F*). Specific probes against native *STE2* and PP7 aptamer-tagged *AGA2* were used to score localization of the *STE2* and *AGA2* mRNAs, respectively (we note that native *AGA2* mRNA is insufficiently long for smFISH). The majority (73.1 ± 1.0%) of *STE2* mRNA spots co-localized with PP7-tagged *AGA2* mRNA spots in untreated cells, while 65.4 ± 0.1% co-localized in cells treated with α-factor (*Figure 2E and F*), and co-localization was observed both in the nucleus and cytoplasm. Thus, mating transperon mRNAs shown to multiplex using biochemical means appear to co-localize before and after nuclear export.

As insertion of the PP7 aptamer sequence before the 3'UTR might potentially alter mRNA fate, we examined the co-localization of endogenously expressed untagged mating mRNAs using smFISH (*Pizzinga et al., 2019*; *Tsanov et al., 2016*). We examined co-localization of the *AGA1* and *STE2* mRNAs and, in parallel, that of the *STE2* and *HSP150* mRNAs in order to show specificity. Overall, we found that 37.9 ± 1.7% (average ± standard error of the mean; SEM) of *AGA1* mRNA co-localized with *STE2* mRNA, while only 9.8 ± 0.6% of *HSP150* mRNA co-localized with *STE2* mRNA (*Figure 3A and B*; representative images are shown in *Figure 3a* and quantified in *Figure 3B*). Together, the

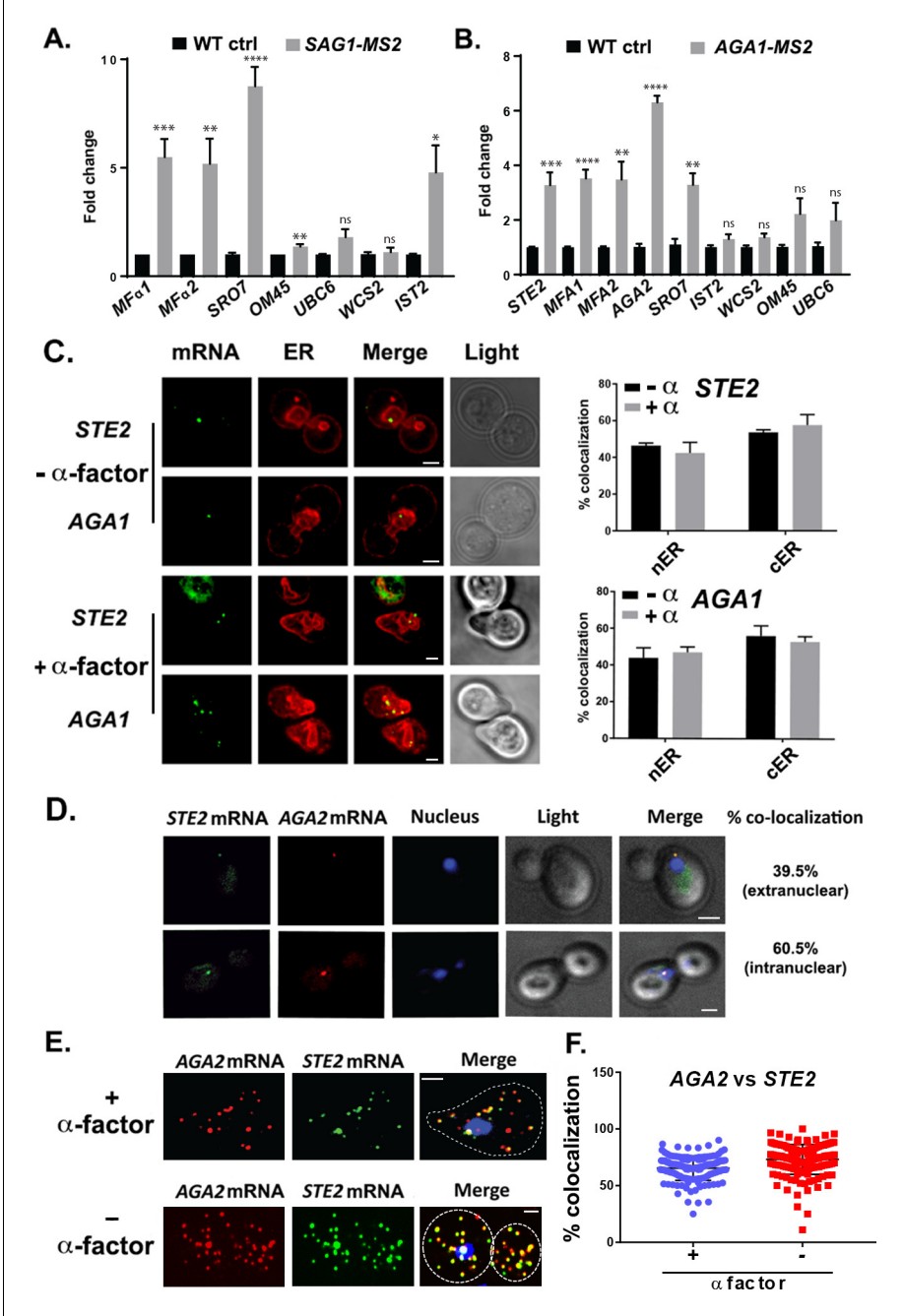

**Figure 2.** mRNAs encoding secreted and membrane proteins involved in mating multiplex in both *MAT*α and *MAT*a cells and localize to the ER. (**A**) A mating messenger ribonucleoprotein (mRNP) complex is also formed in *MAT*a cells. *MAT*a yeast (BY4741) expressing MS2 aptamer-tagged *SAG1* from its genomic locus were grown to mid-log phase (O.D.$_{600}$ = 0.5) and subjected to RaPID followed by qRT-PCR (RaPID-PCR; see 'Materials and methods'). RNA derived from the total cell extract (Input) or biotin-eluated fraction (RaPID) was analyzed by qRT-PCR using primer pairs corresponding to mRNAs expected to possibly multiplex (based on the results shown in *Figure 1*). Three biological repeats were performed and an unpaired t-test was used to compare the *SAG1* pulldown to the WT control pulldown; ****p<0.0001; ***p<0.0005; **p<0.001; *p<0.01. (**B**) Analysis of the mating mRNP complex in *MAT*α cells by RaPID-qPCR. *MAT*α yeast expressing MS2 aptamer-tagged *STE2* from its genomic locus were grown and processed as in (A). RNA derived from the biotin-eluated fraction (RaPID) was analyzed by qRT-PCR using primer pairs corresponding to mRNAs expected to multiplex (based on the results shown in *Figure 1*). Three biological repeats were performed and an unpaired t-test was used to compare the *AGA1* pulldown to the WT control pulldown; ****p<0.0001; ***p<0.0005; **p<0.001. (**C**) Mating mRNAs localize to

*Figure 2 continued on next page*

*Figure 2 continued*

ER in pheromone-treated or -non-treated cells. Wild-type cells expressing MS2 aptamer-tagged *STE2* or *AGA1* mRNAs (as listed) were transformed with single-copy plasmids expressing MS2-CP-GFP(x3) and either Scs2-mCherry (left panel) or Sec63-RFP (right panel). Cells were grown to mid-log phase and either α-factor (5 µM; +α-factor) or dimethyl sulfoxide (DMSO) (-α-factor) was added for 90 min, followed by fixation with 4% paraformaldehyde in a medium containing 3.5% sucrose and prior to imaging by confocal microscopy. Left panel: fluorescent images: *mRNA* – MS2-CP-GFP(x3) labeling; *ER* – Scs2-mCherry labeling; merge – merger of mRNA and ER windows; light – transmitted light. Size bar = 2 µm. Right panel: quantitative analysis of RNA co-localization: nER – nuclear ER; cER – cortical ER. (D) Mating mRNAs *STE2* and *AGA2* co-localize both inside and outside of the nucleus. Yeast expressing MS2 aptamer-tagged *STE2* mRNA and PP7 aptamer-tagged *AGA2* mRNA from its genomic locus were transformed with plasmids expressing MS2-CP-GFP(x3) and PP7-PS-tomato, while the nucleus was stained with 4,6-diamidino2-phenylindole (DAPI). *STE2 mRNA* – MS2-CP-GFP(x3) labeling; *AGA2 mRNA* – PP7-PS-tomato labeling; nucleus – DAPI labeling; merge – merger of mRNA and nucleus windows; light – transmitted light. Size bar = 2 µm. (E) Single-molecule fluorescence in situ hybridization (smFISH) validation of *STE2* and *AGA2* mRNA co-localization. Non-transformed cells from (D) were either treated with α-factor (10 µM) or left untreated and processed for smFISH labeling using non-overlapping FISH probes complementary to the *STE2* mRNA and PP7 aptamers, prior to labeling with DAPI. The representative image shown is from a single focal plane. *AGA2* – Cy3 labeling; *STE2* – Alexa488 labeling; merge – merged Cy3/Alexa488 windows with DAPI staining. Size bar = 2 µm. (F) Scatter plot of co-localization data from (E). Co-localization was assessed using the FISHquant algorithm (see 'Materials and methods'). Black lines indicate the avg. ± SEM for each sample. Each data point represents a single cell.

The online version of this article includes the following figure supplement(s) for figure 2:

**Figure supplement 1.** *MFA1* and *MFA2* mRNAs localize to the cell body.

multiplexing and imaging results suggest that mating pathway mRNAs form particulate mRNP complexes.

## A histone H4 paralog binds to mating mRNP RNAs

To determine how the mating mRNP particle assembles, we performed RaPID-MS using the *MFA1*, *MFA2*, and *STE2* mRNAs either bearing or lacking their 3'UTRs, along with an unrelated control RNA, *ASH1*, as target RNAs. We identified a band of ~150 kDa that was present in all lanes upon silver staining, except for the lanes corresponding to *STE2* mRNA lacking its 3'UTR and *ASH1* mRNA (*Figure 3—figure supplement 1*). Sequencing revealed that this band contained a core histone H4 paralog, Hhf1. Although the predicted molecular mass of Hhf1 is ~11 kDa, the likelihood exists that it was crosslinked with another protein. Indeed, other crosslinked products were found in this band although none were shown to be connected to mRNAs of the mating mRNP (data not shown).

To verify whether Hhf1 alone can precipitate the mating mRNP particle, we performed immunoprecipitation (IP) using HA-tagged Hhf1 and examined the precipitates for the presence of mating mRNAs. First, we found that HA-Hhf1 could precipitate either native or MS2 aptamer-tagged *STE2*, along with native *MFA1* or *MFA2* mRNA, but not *EXO70*, *ASH1*, *SRO7*, or *UBC6* mRNA by RT-PCR (*Figure 3C*). Thus, we could verify a specific interaction between Hhf1 and RNAs of the mating particle and, in addition, show that HA-Hhf1 migrates at around 12 kDa in SDS-PAGE gels (*Figure 3D*), as predicted. Next, we validated these results in HA-Hhf1 pulldowns using qRT-PCR, whereby we could detect native *STE2*, *MFA1*, and *MFA2*, but not *SRO7* or *WSC2*, in the precipitates (*Figure 3E*). A similar experiment was performed with *MATα* cells expressing HA-Hhf1 that identified both *MFα1* and *MFα2*, but not *SRO7* or *WSC2* (n = 2 experiments; data not shown).

## Histone H4 depletion affects mating and co-localization of mRNA

Although histones are principally known for their DNA-binding functions in nucleosome assembly (*Wu and Grunstein, 2000*), we determined whether Hhf1 or its paralog, Hhf2, plays a role in mating RNP assembly and mating. First, we examined whether deletions in H4 or other histone genes affect mating. We performed quantitative mating assays by crossing WT yeast against single gene mutations in the H2A, H2B, H3, and H4 paralog pairs. We found that only deletions in H4 (*hhf1Δ* or *hhf2Δ*) led to a significant decrease (~50–60%) in mating (*Figure 4A*). Since histone paralog pairs are essential, we created a conditional allele of *HHF1* at its genomic locus by adding an auxin-induced degron sequence at the 5' end (*HA-AID-HHF1*) along with an HA epitope, in *hhf2Δ* cells. This allele

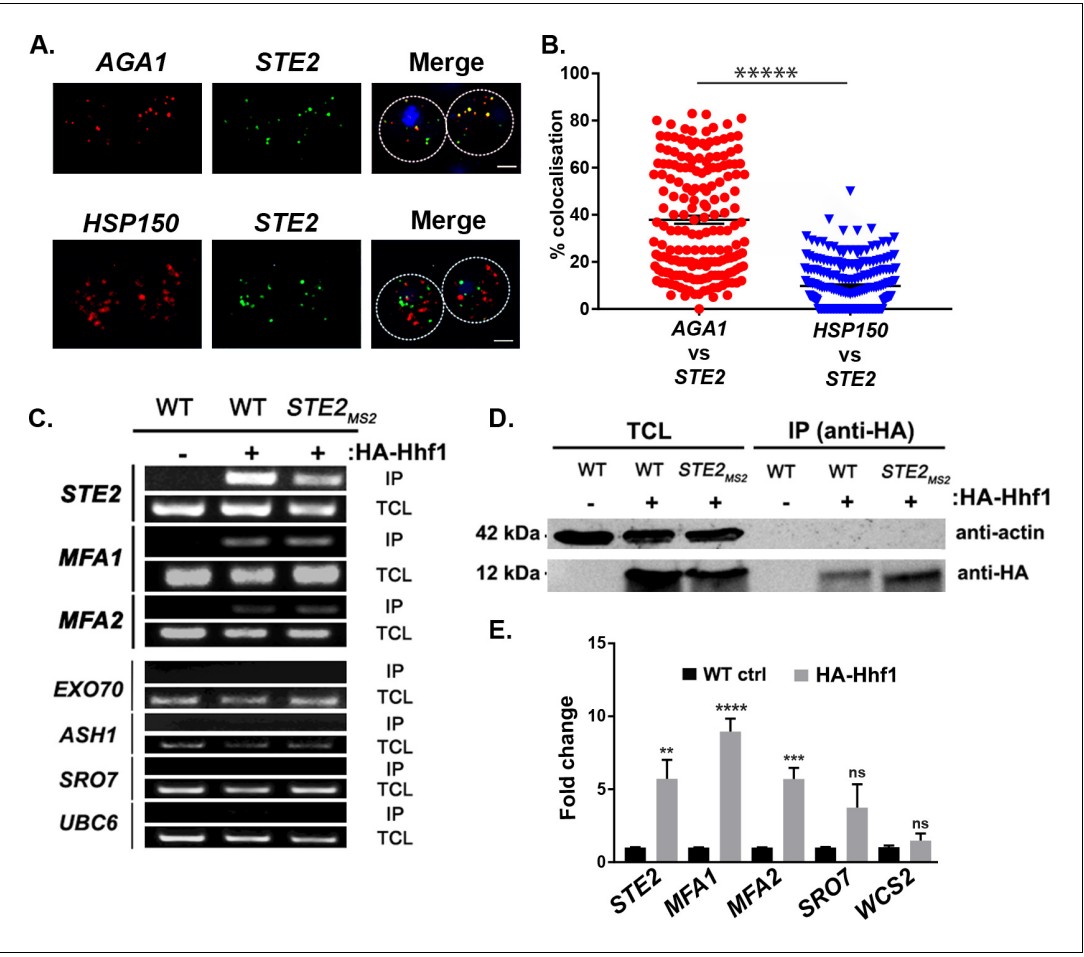

**Figure 3.** Mating pathway mRNAs co-localize and are co-precipitated by HA-Hhf1. (**A**) Endogenous *STE2* and *AGA1* mRNAs co-localize. Wild-type (WT) BY4741 cells were processed for single-molecule fluorescence in situ hybridization (smFISH) labeling using designated FISH probes complementary to the *STE2* and *AGA1* mRNAs, and *STE2* and *HSP150* mRNAs, prior to labeling with 4,6-diamidino2-phenylindole (DAPI). Dotted lines are representative of the cell outline observed in brightfield micrographs. *AGA1* – Cy3 labeling; *STE2* – Alexa488 labeling; *HSP150* – Cy5 labeling; merge – merged Cy3/Cy5 and Alexa-488 windows with DAPI staining. Size bar = 2 μm. The representative image shown is from a single focal plane. (**B**) Scatter plot of co-localization data from (A). Co-localization was assessed using the FISHquant algorithm (see 'Materials and methods'). Black lines indicate the average $\pm$ SEM for each sample. Each data point represents a single cell. ***** indicates p-value<0.00001. (**C**) HA-Hhf1 precipitates mRNAs encoding secreted mating pathway components. *MAT*a WT cells and cells expressing MS2 aptamer-tagged *STE2* mRNA from its genomic locus were transformed with a 2 μ plasmid expressing *HA-HHF1* (+HA-Hhf1). WT cells were also transformed with an empty vector (–HA-Hhf1) as control. Cells were grown to mid-log phase, lysed to yield the total cell lysate (TCL), and subjected to immunoprecipitation (IP) with anti-HA antibodies. Following IP, the RNA was extracted, DNase-treated, and reverse-transcribed. Transcripts were amplified using specific primers to the genes listed. A representative ethidium-stained 1% agarose gel of electrophoresed samples is shown; n = 3 experiments. (**D**) HA-Hhf1 migrates as a 12 kDa protein in SDS-PAGE gels. Aliquots (5% of total) of the TCL and IP fractions were resolved on 15% SDS-PAGE gels and transferred to nitrocellulose membranes for detection with anti-HA and actin antibodies to verify the presence of HA-tagged Hhf1 in both fractions. (**E**) HA-Hhf1 precipitates the mating mRNAs. Same as in (C), except the presence of mating mRNAs was analyzed by qRT-PCR using primer pairs corresponding to mRNAs expected to co-purify with HA-Hhf1, as well as with different controls.

The online version of this article includes the following figure supplement(s) for figure 3:

**Figure supplement 1.** Identification of an RNA-binding protein that binds to the 3'UTR of the mating mRNAs.

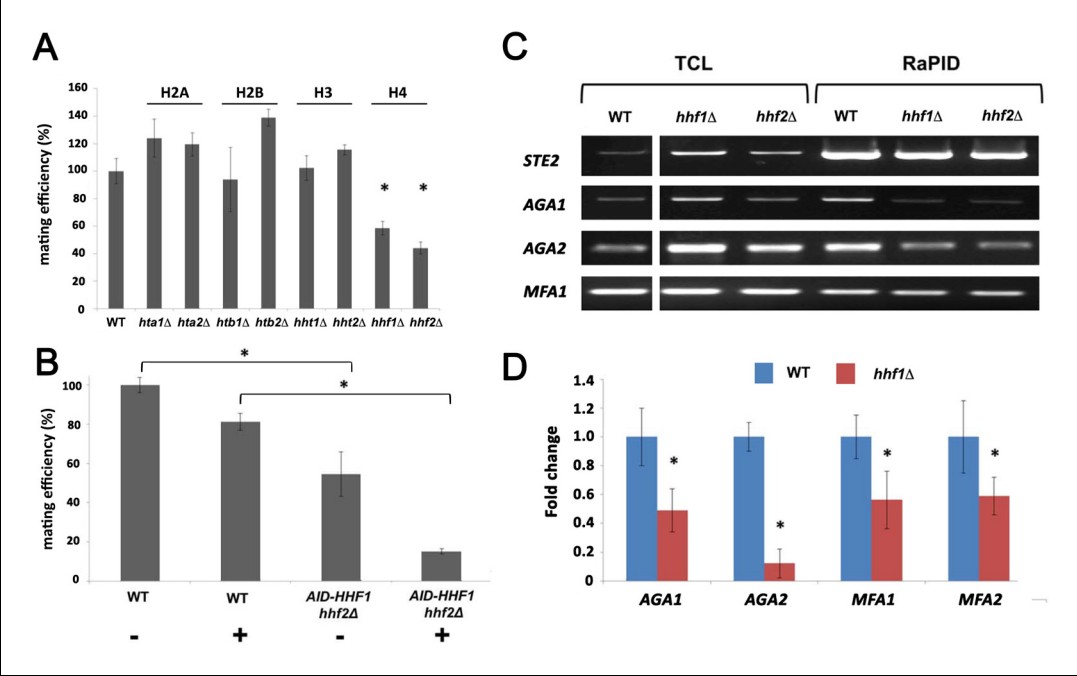

**Figure 4.** Histone H4 function is required for mating mRNP assembly and mating. (**A**) Histone H4 alone is required for mating. *MAT*a yeast (BY4741) bearing mutations in H2A (*hta1Δ, hta2Δ*), H2B (*htb1Δ, htb2Δ*), H3 (*hht1Δ, hht2Δ*), and H4 (*hhf1Δ, hhf2Δ*) were grown to mid-log phase, crossed against *MAT*α wild-type (WT) cells, and mating was scored using the quantitative mating assay. n = 3 experiments; *p-value<0.05 between mutant and WT. (**B**) Combined *hhf2Δ* and *AID-HHF1* alleles result in a near-complete block in mating upon auxin induction. *MAT*a WT and *hhf2Δ AID-HHF1* cells were grown to mid-log phase, treated either with or without auxin (3-IAA; 4 mM) for 4 hr, and examined for mating using the quantitative mating assay. n = 3 experiments; p-value<0.05 between mutant and WT. (**C**) Deletions in H4 histones result in defects in mating mRNP assembly – PCR analysis. *MAT*a WT, *hhf1Δ*, and *hhf2Δ* cells expressing MS2 aptamer-tagged *STE2* from the genome were grown to mid-log phase, processed for RaPID-PCR, and the extracted RNA analyzed using primers against the listed mating mRNAs. A representative ethidium-stained 1% agarose gel of electrophoresed samples is shown; n = 3 experiments. (**D**) Deletions in an H4 histone result in defects in mating mRNP assembly – qRT-PCR analysis. Same as in (C), except that qRT-PCR was performed instead of PCR on WT and *hhf1Δ* cells expressing MS2 aptamer-tagged *STE2* from the genome. Three experiments were performed and gave similar results; p-values<0.05 between mutant and WT cells.

The online version of this article includes the following figure supplement(s) for figure 4:

**Figure supplement 1.** Conditional removal of both histone H4 paralogs leads to synthetic lethality over extended time.

had no effect upon growth in the absence of auxin (3-indole acetic acid; 3-IAA), but led to lethality upon long-term treatment in the presence of 3-IAA (4 mM) (***Figure 4—figure supplement 1A***). Growth of the cells in presence of 3-IAA resulted in a full depletion of Hhf1 within 4 hr (***Figure 4—figure supplement 1B***), and cells remained viable after this short-term treatment (***Figure 4—figure supplement 1C***). When examined for mating after pre-treatment with 3-IAA (4 mM; 4 hr), we observed a >80% inhibition in mating (***Figure 4B***). Thus, the depletion of histone H4 strongly affects yeast cell mating, even under conditions where viability is maintained.

To check whether the depletion of H4 results in changes in the mating RNA levels, we examined *STE2* and *AGA1* transcript levels using qRT-PCR in WT and *AID-HHF1 hhf2Δ* cells either with or without added 3-IAA and/or α-factor. In the cases of *STE2* and *AGA1*, we observed similar increases in mRNA transcript levels in either cell type upon α-factor addition (***Figure 4—figure supplement 1D***) with or without added 3-IAA. Thus, auxin treatment does not abolish the increase in mating mRNA levels observed upon pheromone addition. In fact, we noted that 3-IAA addition alone had a small stimulatory effect on *AGA1* RNA levels (***Figure 4—figure supplement 1D***).

We analyzed the co-localization of *AGA1* and *STE2* mRNAs in WT cells and *AID-HHF1 hhf2Δ* cells either with or without the addition of 3-IAA (*Figure 4—figure supplement 1E*). While WT cells gave 37.9 ± 1.7% co-localization, this was reduced to 15.9 ± 1.0 and 14.0 ± 0.9% (avg. ± SEM) either without or with 3-IAA addition, respectively. We note that the addition of auxin resulted in a greater number of *AID-HHF1 hhf2Δ* cells that showed no co-localization.

## Deletion of histone H4 alleles affects mating transperon assembly

As the depletion of H4 greatly lessens the mating propensity of yeast, we examined whether this is a consequence of altered mRNP particle assembly. We first performed RaPID-PCR on *STE2* mRNA derived from *hhf1Δ* and *hhf2Δ* cells, and measured the co-precipitation of other mating pathway mRNAs. We found decreased levels of *AGA1, AGA2,* and, to a lesser degree, *MFA1* mRNA in the *hhfΔ* deletion strains (*Figure 4C*). This result was verified using RaPID-qPCR where the levels of bound *AGA1, MFA1,* and *MFA2* mRNAs declined by ~50%, while those of *AGA2* declined even more (*Figure 4D*). The results indicate that the reduction in H4 amounts results in defects in mating mRNP complex formation, which may account for the loss in mating efficiency (*Figure 4B*).

## Histone acetyl transferases and histone H4 acetylation are required for mating

The N-terminus of histones is extensively modified by methylation, acetylation, and phosphorylation (*Allis and Jenuwein, 2016*). We examined the mating efficiency of yeast upon the deletion of three histone acetyl transferase genes (HATs; e.g., *HAT1, HAT2,* and *SAS2*) (*Kurdistani and Grunstein, 2003*) that were found to be physically and genetically linked with the *HHF1* gene and its product, according to the Saccharomyces Genome Database (SGD). We found that the deletion of *HAT1* or *SAS2* had the same reduced efficiency as observed with *hhf1Δ* cells (*Figure 5A*). This implies that the acetylated state of H4 may be necessary for function.

To verify this, we mutated all N-terminal Hhf1 lysine residues to arginines (e.g., K6R, K9R, K13R, K17R, K32R; K-to-R mutant) to abolish acetylation by HATs, or to glutamines (e.g., K6Q, K9Q, K13Q, K17Q, K32Q; K-to-Q mutant), to mimic the acetylated state. We mated *MAT*a cells expressing these mutants with WT *MAT*α cells and assessed the mating efficiency. Overexpression of WT *HHF1* caused a >2-fold increase in mating efficiency (*Figure 5B*), while overexpression of the K-to-R mutant abolished it. Correspondingly, overexpression of the K-to-Q mutant yielded the same two-fold increase in mating efficiency as observed upon WT *HHF1* overexpression (*Figure 5B*). Since the K-to-R mutation appeared to be non-functional, we expressed this mutant from the genome and examined mating efficiency. We found that the expression of this sole copy of *HHF1* led to a 50% decrease in mating (*Figure 5C*), but without lowering *HHF1* mRNA expression (e.g., level of *HHF1*K-to-R=1.7 ± 0.6-fold over *HHF1* in WT cells [avg. ± SEM, n = 3]). While we cannot rule out the possibility that the K-to-R mutation results in protein instability leading to reduced mating efficiency, nevertheless, it appears that histone H4 acetylation is important for the mating process.

## Histone H4 and Scp160 work cooperatively to confer mating

A previous work from the lab identified Scp160 as an ER-localized RBP involved in the delivery of certain mPOLs (e.g., *SRO7*) and mRNAs that confer the internal MAP kinase (MAPK) mating signal (e.g., *FUS3, KAR3, STE7*) to the shmoo tip upon pheromone treatment (*Gelin-Licht et al., 2012*). Deletion of *SCP160* was shown to prevent the polarized trafficking of these mRNAs on cER, resulting in a loss in chemotropism, a 60–70% decrease in mating in heterozygous crosses, and a >98% decrease in homozygous crosses. As mating transperon RNAs also localize to ER (*Figure 2C*), although not asymmetrically localized to the bud or shmoo tips, we examined whether Hhf1 and Scp160 work together or separately on distinct signaling paths. To do this, we examined whether deletions of either *HHF1* or *SCP160* in the separate mating partners or together, as combined mutations in one of the mating partners, had additive/synergistic deleterious effects upon mating. We crossed *MAT*α *hhf1Δ* cells against *MAT*a *scp160Δ* cells and observed a block in mating similar to that of *hhf1Δ* x WT crosses (*Figure 5D*), indicating that there was no additive defect when single gene mutations are present in the separate mating partners. However, when *hhf1Δ scp160Δ* double mutants are crossed against WT cells, we observed a complete block in mating (i.e., sterility), reminiscent of a *scp160Δ* homozygous cross (*Gelin-Licht et al., 2012*). This indicates that *HHF1* and

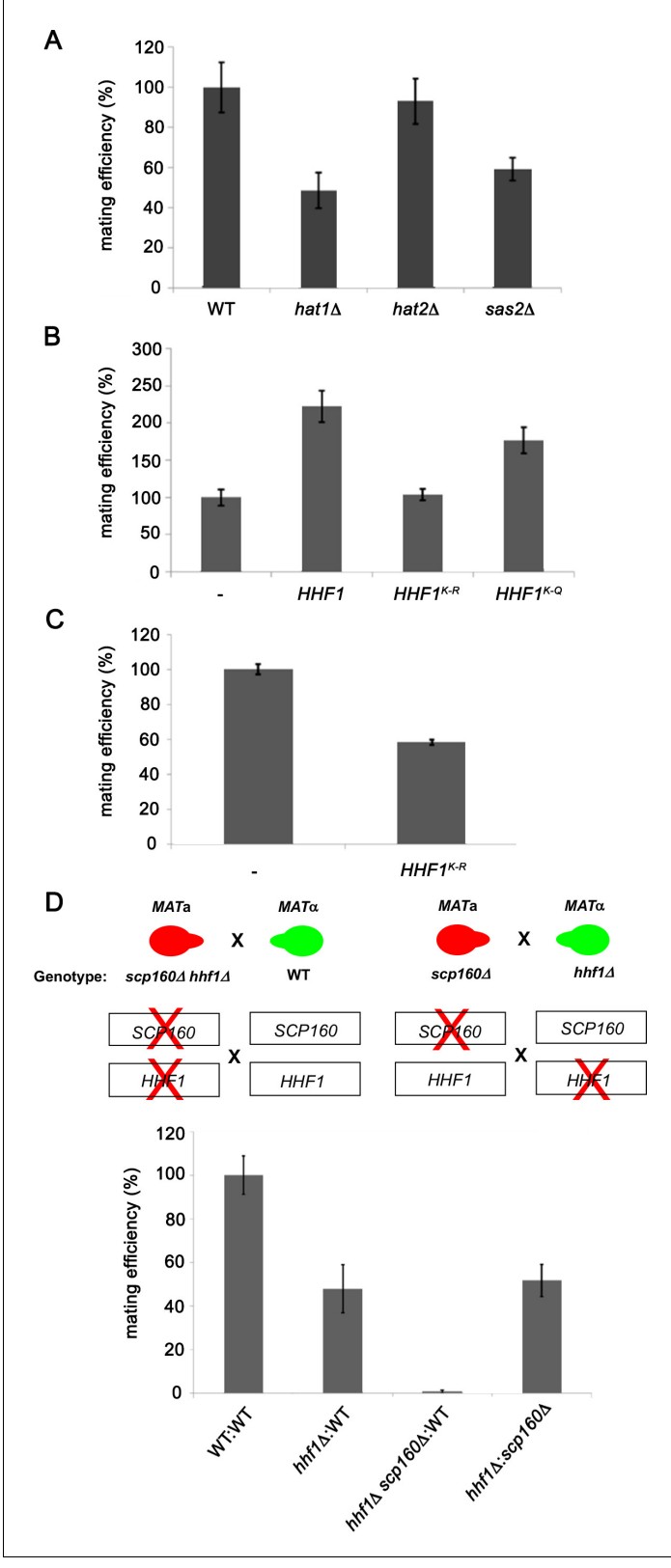

**Figure 5.** Histone H4 acetylation is required for mating. (**A**) Histone acetyltransferases are required for mating. Wild-type (WT; BY4741) and yeast lacking known acetyltransferase genes (*hat1Δ, hat2Δ,* and *sas2Δ*) were grown to mid-log phase (O.D.$_{600}$ = 0.5) and examined for mating with WT *MATα* cells (BY4742). Three quantitative mating experiments were performed and gave similar results; p-values<0.05 between mutants and WT cells. (**B**) Mutations

*Figure 5 continued on next page*

*Figure 5 continued*

that either mimic or block N-terminal histone H4 acetylation increase or decrease mating efficiency, respectively. *MAT*a WT (BY4741) cells alone (-) or overexpressing *HHF1* (*HHF1*), along with cells overexpressing *HHF1* bearing K-to-R (*HHF1*$^{K-R}$) or K-to-Q (*HHF1*$^{K-Q}$) mutations from a 2 um plasmid, were grown to mid-log phase and examined for mating with *MAT*α cells. Three experiments were performed; p-values<0.05 between mutants and WT cells. (C) Expression of the K-to-R mutation in *HHF1* in the genome inhibits mating comparative to the deletion of *HHF1*. *MAT*a WT (BY4741) cells alone (–) or expressing *HHF1* bearing the K-to-R (*HHF1*$^{K-R}$) mutation from the genome were grown to mid-log phase and examined for mating with *MAT*α cells. Three experiments were performed; p<0.05. (D) *HHF1* and *SCP160* are not epistatic, and *hhf1Δ scp160Δ* double mutants are sterile. Crosses between WT, *hhf1Δ*, or *hhf1Δ scp160 MAT*a cells and *MAT*α WT cells, along with an *hhf1Δ* and *scp160Δ* cross, were performed. Illustrated are the *hhf1Δ scp160* x WT cross (upper left side) and the *hhf1Δ* x *scp160Δ* cross (upper right side). Three experiments were performed; p-values<0.05 between crosses.

---

*SCP160* contribute more or less equally to the mating process (in terms of conferring mating efficiency) and act in a non-epistatic fashion.

## *MAT*a mating particle RNAs have a conserved sequence motif important for mRNA assembly, pheromone responsiveness, and mating

As mating transperon mRNAs multiplex with the help of histone H4, we examined their sequences for recognizable motifs by bioinformatic analysis (e.g., MEME Suite; MEME) (*Bailey et al., 2009*). Although we anticipated that the H4 interaction site might reside (at least in part) within the 3'UTR, based on the RaPID-MS experiment (*Figure 3—figure supplement 1*), we identified a single motif of 47 nucleotides present in the coding region of all five *MAT*a messages examined (*Figure 6A*). To determine whether this motif facilitates RNA multiplexing and mating, we created two mutated versions in *AGA2* (e.g., a short 27-nucleotide version [*AGA2 27-MUT*] and a full-length 47-nucleotide version [*AGA2 47-MUT*]), which substituted certain conserved residues within the motif with synonymous mutations that did not alter the amino acid sequence. Using RNA-fold (http://rna.tbi.univie.ac.at/cgi-bin/RNAWebSuite/RNAfold.cgi), we found that the structure of *AGA2 47-MUT* was distorted, as compared to *AGA2 27-MUT* and the native *AGA2* motif (*Figure 6—figure supplement 1A*). When substituted for the native motif in *AGA2*, we found that *AGA2 47-MUT*, but not *AGA2 27-MUT*, led to a loss of *AGA2* mRNA assembly into the mating mRNP particle precipitated using *STE2* as the pulldown mRNA (*Figure 6B*). We note that the *AGA2 47-MUT* mutation did not alter gene expression (e.g., *AGA2 47-MUT* expression was $1.2 \pm 0.2$-fold over *AGA2* in WT cells [avg. ± SEM, n = 3]). However, an increase in *AGA2 27-MUT* gene expression (e.g., $2.7 \pm 0.6$-fold; avg. ± SEM, n = 3) was observed, which might explain why it had less significant effects upon multiplexing. Similarly, the deletion of *AGA2* itself also led to defects in mating mRNP assembly (*Figure 6—figure supplement 1B*), and either motif mutation or gene deletion resulted in significant defects in the ability of cells to mate with WT yeast (*Figure 6C*). These results suggest that motif presence plays an important role in both mating transperon formation and mating.

Bioinformatic analyses revealed that this motif was not present in the mating mRNAs expressed in *MAT*α cells; however, two distinct non-overlapping motifs of 45 and 25 bases could be identified in the *MF*α*1*, *MFA*α*2*, *STE3*, and *SAG1* mRNAs (*Figure 6—figure supplement 1C*). Thus, both mating types appear to utilize different *cis* elements in their RNAs.

To determine whether particle assembly is ultimately important for both cellular responsiveness to the external pheromone signal (i.e., pheromone secreted by cells of the opposite mating type) and the secretion of pheromone and ability to induce a mating response in WT cells, we crossed *MAT*a cells expressing these *AGA2* mutants with GFP-labeled WT *MAT*α cells and assessed the shmooing efficiency of both cell types (*Figure 6—figure supplement 2A*). In parallel, we examined whether deletions in *HHF1*, *HHF2*, and *AGA1* also affected shmooing or the induction of shmoo formation in WT cells of the opposite mating type. We observed reduced shmoo formation for both the wild-type *MAT*α and *MAT*a deletion strains in the non-isogenic crosses, as compared to the isogenic WT *MAT*α and WT *MAT*a cross (*Figure 6—figure supplement 2A*). The reduction in shmoo formation (i.e., pheromone responsiveness) was detectable whether either histone H4 paralog was deleted or whether *AGA1* or *AGA2* was mutated. We note that *AGA2 27* mutant had the least effect upon shmooing, whereas the histone deletions were somewhat more effective.

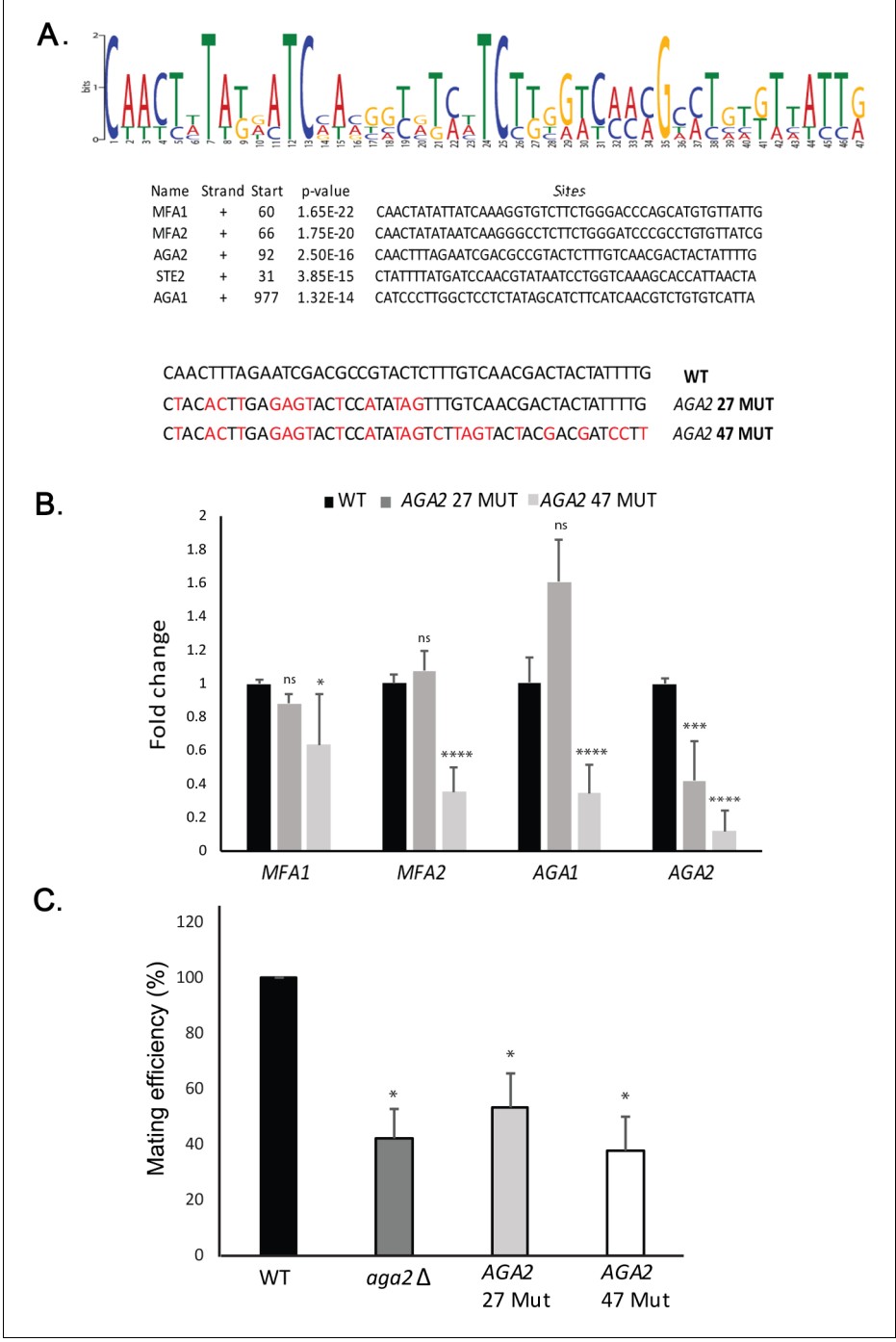

**Figure 6.** A motif in the mating mRNA coding region facilitates mRNP assembly and mating. (**A**) A conserved motif is common to all five *MAT*a mRNAs. MEME-ChIP analysis of the sequences of all five *MAT*a mRNAs was performed and revealed a 47-nucleotide consensus motif in the coding regions, shown schematically as a sequence logo based on nucleotide representation. Also shown are two forms (*AGA2 27-MUT* and *AGA2 47-MUT*) in which the motif was altered within the coding region to remove conserved nucleotides without altering the amino acid sequence. (**B**) Mutation of the consensus motif in *AGA2* leads to an inhibition in messenger ribonucleoprotein (mRNP) particle assembly. MS2 aptamer-tagged *STE2* cells bearing the *aga2Δ* deletion were transformed with a single-copy plasmid expressing *AGA2* or either of the two *AGA2^mut^* forms. Next, WT, *AGA2 27-MUT*, and *AGA2 47-MUT* cells expressing MS2-tagged *STE2* were subjected to RaPID-qPCR with primers against *AGA1, AGA2* (note: not within mutated region), *MFA1*, and *MFA2*. The histogram indicates the levels of precipitated mating mRNAs from wild-type (WT) (black) or *AGA2 27-MUT* and *AGA2 47-MUT* (dark and light grey,

*Figure 6 continued on next page*

*Figure 6 continued*

respectively) cells after normalization for the level of *STE2* mRNA pulldown. Three biological repeats were performed and an unpaired t-test was performed to compare *AGA2 27-MUT* and *AGA2 47-MUT* with WT; ****p<0.0001; ***p<0.0005; *p<0.01. (C) Deletion of *AGA2* or mutation of the consensus motif alters mating efficiency. *MATa* WT cells (WT; BY4741), *AGA2 27-MUT* or *AGA2 47-MUT* mutant cells were crossed against WT *MATα* cells and quantitative mating was assessed. Three biological repeats were performed and an unpaired t-test was performed to compare *AGA2 27-MUT* and *AGA2 47-MUT* with WT; *p<0.05.

The online version of this article includes the following figure supplement(s) for figure 6:

**Figure supplement 1.** Structure of native and mutated *AGA2* mating motifs, effect of *AGA2* deletion on mating mRNP assembly, and identification of consensus motifs in *MATα* mating mRNAs.

**Figure supplement 2.** Mutations in *AGA2* (in *MATa* cells) or histone H4 alter both mating partners' responsiveness to pheromone, and P-body components are not essential for mating.

These results suggest that defects in mRNP assembly affect not only the mutant cell's ability to secrete pheromone and, thus, influence its WT mating partner but also the mutant cell's ability to respond to pheromone secreted by its WT mating partner. Therefore, mating transperon assembly likely impacts upon both pheromone synthesis/secretion as well as pheromone responsiveness.

## P-body elements are not required for yeast cell mating

Our results show that mating mRNAs localize to ER in the cell body of both pheromone-treated and -untreated cells (*Figure 2C*) and, in the case of *MFA2*, do not associate with P-bodies under conditions of endogenous gene expression (*Haimovich et al., 2016*). In contrast, it was proposed earlier that mating mRNAs, like *MFA2*, localize to P-bodies in the shmoo tip and that P-body components are necessary for transmission of the mating signal (*Aronov et al., 2015*).

To re-examine the involvement of P-bodies in the mating process, we created deletions in *DHH1* and *PAT1* (e.g., *dhh1Δ*, *pat1Δ*, and *dhh1Δ pat1Δ*), which encode proteins necessary for P-body formation, and measured the ability of these mutants to mate with WT cells using the quantitative mating assay. However, unlike the previously published results (*Aronov et al., 2015*), we found that these same deletion mutants had no significant inhibitory effect upon mating (*Figure 6—figure supplement 2B*).

To further validate our findings, we analyzed the co-localization of endogenously expressed *STE2* mRNA with a P-body marker, Dhh1-mCherry, under growth conditions that induce P-body formation (i.e., 10 min in a no-glucose medium) (*Figure 6—figure supplement 2C*). Under non-inducing conditions (i.e., 2% glucose), we observed few cells with Dhh1-mCherry puncta (2.3 ± 0.2%; avg. ± SEM) and little to no co-localization between the multiple mRNAs detected and the puncta observed. In contrast, all cells formed discrete Dhh1-mCherry-labeled puncta when shifted to a no-glucose medium and we observed 40.9 ± 5.2% (avg. ± SEM) co-localization with *STE2* mRNA. Thus, mating pathway mRNAs do not co-localize with a P-body marker under normal growth and mating conditions. However, upon exposure to no-glucose stress, which blocks cell mating, these RNAs can reside within P-bodies.

## Intergenic association between the *STE2* and *AGA2* genes

Since we observed *STE2* and *AGA2* mRNA co-localization both before and after nuclear export (*Figure 2D and F*), we hypothesized that their genes (located on ChrVI and ChrVII, respectively) might undergo interchromosomal interactions to facilitate RNA multiplexing upon transcription. To examine the possibility of allelic coupling, we performed chromatin conformation capture (3C) (*Lieberman-Aiden et al., 2009*), using multiple controls. First, we confirmed using qPCR that the chromatin was restriction enzyme-digested by >95%; second, we used a non-ligated control as a measure of background for ligation-dependent 3C signals; third, we used the *ASH1* gene as a negative control, as its mRNA does not associate with the mating mRNP particle; and fourth, to show allele-specific coupling, we analyzed interactions between the *GET1* and *DDI2* genes, which are located in the same chromosomes as *STE2* and *AGA2*, respectively (*Figure 7A and B*). Importantly, we observed three interactions between *STE2* and *AGA2*, consisting of ORF-ORF, ORF-5'UTR, and 3'UTR-3'UTR (*Figure 7—figure supplement 1A*), which were confirmed by DNA sequencing (*Figure 7—figure supplement 1B*). These interactions were specific, as they were not observed

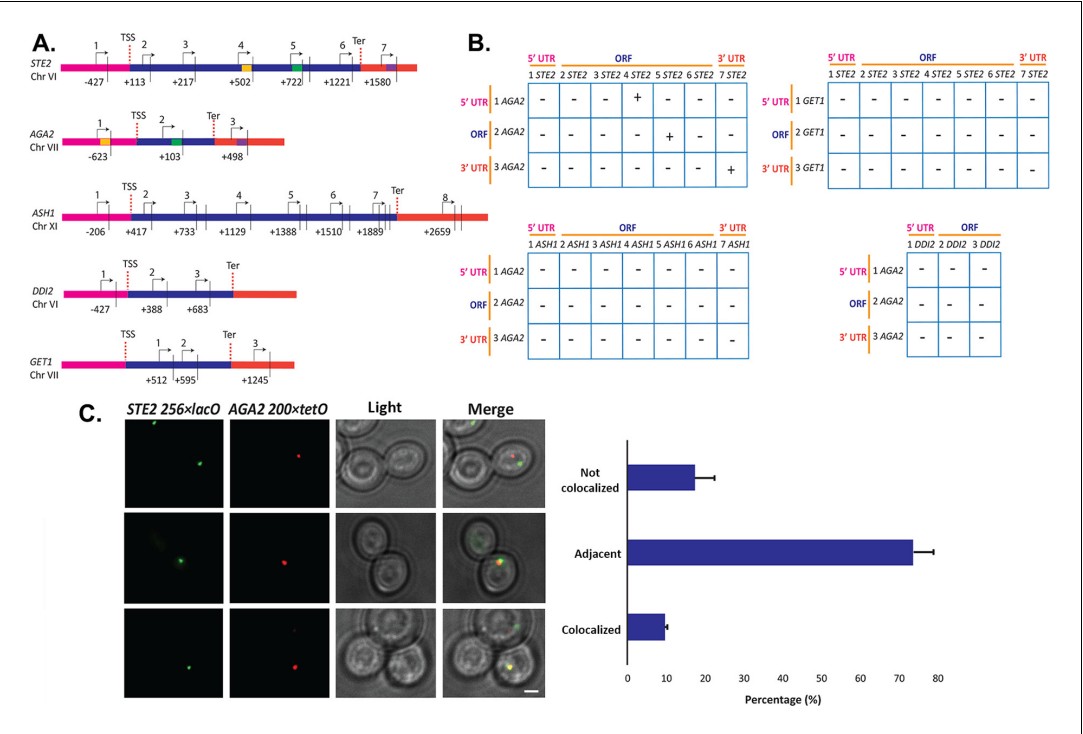

**Figure 7.** Intergenic association of *STE2* and *AGA2* genes. (A) Schematic of the *STE2, AGA2, ASH1, GET1,* and *DDI2* genes and the oligonucleotides used for their amplification from 3C DNA samples. Coordinates correspond to *Taq*I sites (shown as vertical black bars); site numbering is relative to ATG (+1). Forward primers used for 3C analysis were sense-strand identical (arrows) and positioned proximal to *Taq*I sites as indicated. Primers are numbered to distinguish the pairs used in the PCR reactions shown in (B). 5′UTRs, ORFs, and 3′UTRs are color-coded, as indicated. Transcription start sites (TSS) and termination sites (Ter) are indicated. (B) Matrix summarizing the intergenic association of represented genes as determined by 3C-PCR. Primer pairs corresponding to the different genes listed in (A) were used in PCR reactions. '+' indicates PCR amplification and the interaction between genes. '−' indicates no amplification. See *Figure 7—figure supplement 1A* for examples of both. 5′UTRs, ORFs, and 3′UTRs are indicated by broken orange lines. (C) Live cell fluorescence microscopy of *AGA2−224 × tetR* and *STE2−256 × lacO* yeast grown to mid-log phase prior to imaging. Left panel: *STE2−256 × lacO* gene is labeled with GFP-lacI; *AGA2−224 × tetR* is labeled with tetR-tdtomato; merge – merger of *STE2* and *AGA2* windows; light – transmitted light. Size bar = 2 µm. Right panel: histogram of the data from three biological replicas (avg. ± std.dev.); co-localized – fully overlapping signals; adjacent – partially overlapping signals; not co-localized – no overlap between signals. The representative image shown is from a single focal plane.

The online version of this article includes the following figure supplement(s) for figure 7:

**Figure supplement 1.** Intergenic association of *STE2* and *AGA2* genes.

between *STE2* and *GET1*, or *AGA2* and *DDI2*, or between *ASH1* and *AGA2*, for example (see *Figure 7—figure supplement 1A*, for example of latter).

To confirm the interaction between the *STE2* and *AGA2* genes, we created yeast bearing *AGA2*-tagged upstream of the 5′UTR with 224 tetracycline repressor repeats (*224 × tetR*) and *STE2*-tagged upstream of its 5′UTR with 256 lac operator repeats (*256 × lacO*) and performed live fluorescence imaging using co-expressed tetR-tdTomato and GFP-tagged lacI, respectively. We observed 9.5% fully co-localized signals (i.e., 100% overlapping) and 73.6% partially co-localized signals (10–25% overlapping), where both loci appear adjacent to each other (*Figure 7C*). In contrast, only 17.3% of the loci were not closely associated. As a control, we performed live imaging of *AGA2 224 × TetR* and the *ASH1* gene tagged with *256 × LacO* repeats and the abovementioned fluorescent reporters (*Figure 7—figure supplement 1C and D*), and observed far less locus co-localization or adjacent loci (e.g., 2.3 and 27.2%, respectively), as compared to *AGA2 224 × TetR* and *STE2 256 × LacO* (*Figure 7C*). These results suggest that the *AGA2* and *STE2* loci exist in close association within the nucleus. Next, to determine if the allele interaction is transcription-mediated, we treated *AGA2 224 × TetR-* and *STE2 256 × LacO*-tagged cells with 1,10 phenanthroline, which inhibits general transcription. We observed a modest increase (40–50%) in non-co-localized *AGA2* and *STE2* alleles

(*Figure 7—figure supplement 1E*), which suggests that allele apposition may be mediated by transcription. Likewise, the deletion of *HHF1* alone reduced the level of allelic coupling in *AGA2 224 × TetR* and *STE2 256 × LacO* cells (*Figure 7—figure supplement 1F*), which could indicate a structural role for H4 in leading to RNA multiplexing.

## mRNAs encoding heat shock proteins multiplex to form a heat shock RNP particle

*Chowdhary et al., 2019* reported an intergenic association of HSP genes in yeast undergoing heat shock. Based on this, we predicted that HSP mRNAs might also undergo multiplexing due to HSP gene allelic coupling. Thus, we tagged *HSP104* with the MS2 aptamer and performed RaPID to identify cohort HSP mRNAs in the pulldowns. Importantly, the mRNAs of HSP genes previously shown to undergo intergenic interaction (e.g., *TMA10*, *HSP12*, *HSP82*, *SSA2*, and *SSA4*) (*Chowdhary et al., 2019*) were found to co-precipitate both before, but much more after heat shock (*Figure 8A*). We also examined the co-localization of *HSP104* and *SSA2* mRNAs under both non-heat shock and heat

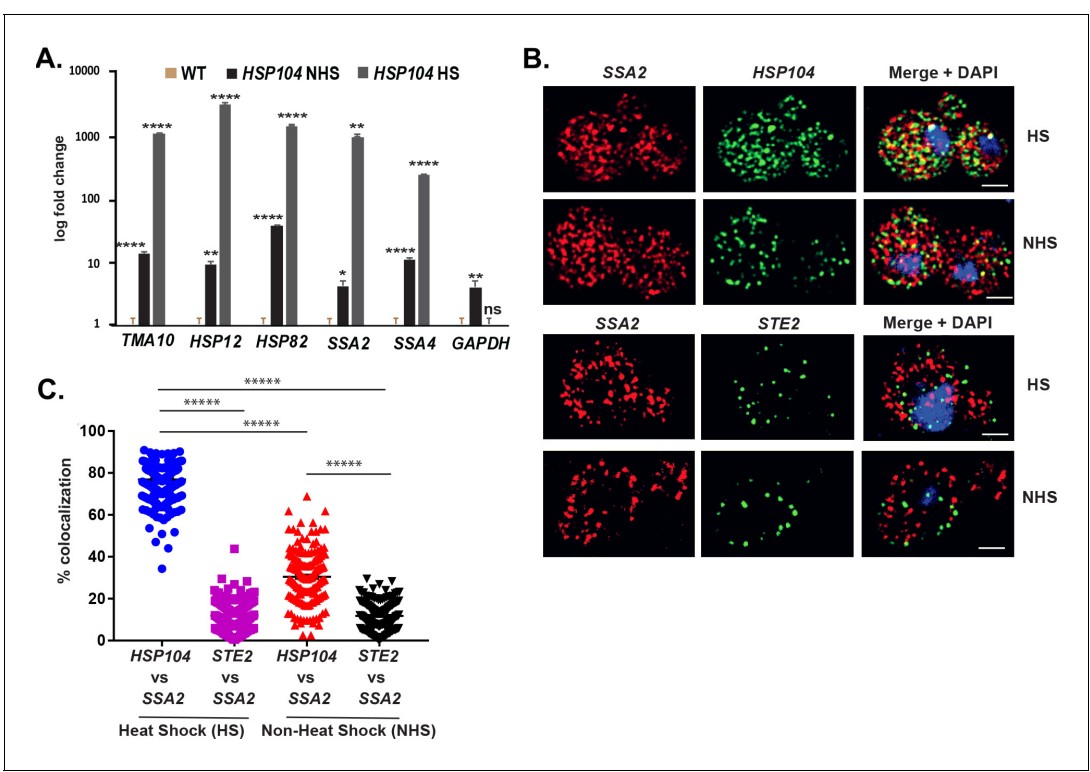

**Figure 8.** mRNAs encoding heat shock proteins undergo enhanced multiplexing upon heat shock. *MAT*a yeast strains (BY4741) expressing MS2 aptamer-tagged *HSP104, UGO1*, or *STE5* from their genomic loci or untagged control (WT) cells were grown to mid-log phase (O.D.$_{600}$ = 0.5) and subjected to RaPID followed by qRT-PCR (RaPID-qRTPCR; see 'Materials and methods'). RNA derived from the total cell extracts or biotin-eluated fractions was analyzed by qRT-PCR using primer pairs corresponding to mRNAs expected to multiplex (see listed genes) or not (e.g., *GAPDH* mRNA). (A) mRNAs encoding heat shock proteins (HSPs) multiplex upon heat shock. Cells expressing MS2 aptamer-tagged *HSP104* were either exposed to heat shock (10 min; 40°C; *HS*) or maintained at 30°C (*NHS*) prior to fixation and RaPID-qRT-PCR. Three biological replicates were performed and an unpaired t-test was used to compare each under non-heat and heat shock conditions with WT; ****p<0.0001; **p<0.001; *p<0.01. (B) mRNAs of HSPs co-localize upon heat shock. Representative single-molecule fluorescence in situ hybridization (smFISH) images of BY4741 WT cells that underwent heat shock (HS) or did not (NHS). BY4741 were processed for smFISH labeling using designated FISH probes complementary to the *HSP104* and *SSA2* and *HSP104* and *STE2* mRNA pairs prior to labeling with DAPI. *SSA2* – Cy3 labeling; *HSP104/STE2* – Alexa488 labeling; merge – merged Cy3 and Alexa-488 windows with DAPI staining. Size bar = 2 µm. The representative image shown is from a single focal plane. (C) Scatter plot of data from (B) showing the proportion of co-localized smFISH foci. Black lines represent average ± SEM distribution. Each data point represents a single cell. *****p-value<0.00001.

The online version of this article includes the following figure supplement(s) for figure 8:

**Figure supplement 1.** Motifs in HSP mRNA coding regions and role of histone H4 in transperon formation.
**Figure supplement 2.** Histone H4 function is required for HSP mRNP assembly.

shock conditions by smFISH (*Figure 8B and C*). Under non-heat shock conditions, we found that 30.4 ± 0.9% of *HSP104* mRNA co-localized with *SSA2* mRNA, whereas this increased to 76.8 ± 0.5% (avg. ± SEM) upon heat shock (*Figure 8B and C*; representative images shown in *Figure 8B* and quantified in *Figure 8C*). As a control, we visualized *STE2* mRNA in co-localization experiments with *SSA2* mRNA. We found similar low levels of *STE2* co-localization with *SSA2* mRNA either with or without heat shock (11.9 ± 0.5% and 11.7 ± 0.4%, respectively) (*Figure 8B and C*). Thus, like mating pathway mRNAs, HSP mRNAs also multiplex into specific transperons, especially during heat shock.

We examined the HSP gene sequences for recognizable motifs and identified three potential motifs (*Figure 8—figure supplement 1A*) present in nearly all HSP cohort genes (*Figure 8—figure supplement 1B*). To determine whether histone H4 is involved in the assembly of the HSP transperon, we performed a growth test for the different histone deletion mutants upon heat shock. While single gene deletions did not affect doubling time after heat shock (*Figure 8—figure supplement 1C*), *AID-HHF1 hhf2Δ* cells treated with 3-IAA showed a longer doubling time after heat shock (at all times examined), as compared to WT cells or untreated *AID-HHF1 hhf2Δ* cells (*Figure 8—figure supplement 1D*). Moreover, we found that *HSP* mRNA (e.g., *TMA10, HSP12, HSP82, SSA2,* and *SSA4*) multiplexing with *HSP104* mRNA was reduced significantly in these cells after heat shock in the presence of 3-IAA (*Figure 8—figure supplement 2A and B*).

We next analyzed *HSP104* and *SSA2* mRNA co-localization in both WT and *AID-HHF1 hhf2Δ* cells, either with or without the addition of 3-IAA (*Figure 8—figure supplement 2C*). Upon heat shock, we observed reduced mRNA co-localization in *AID-HHF1 hhf2Δ* cells either with (51.5 ± 0.9%; avg. ± SEM) or without (55.8 ± 0.9%; avg. ± SEM) the addition of 3-IAA, as compared to WT cells (76.8 ± 0.5%). We note that cells with no co-localized *HSP104* and *SSA2* mRNA were observed only in 3-IAA-treated *AID-HHF1 hhf2Δ* cells. Since changes in RNA expression might affect co-localization, we analyzed *HSP104* and *SSA2* expression using qRT-PCR. Importantly, we did not observe changes in expression without the addition of 3-IAA under either heat shock or non-heat shock conditions. An increase in *HSP104* and *SSA2* gene expression was observed under non-heat shock conditions after the addition of 3-IAA; however, no overall differences were observed between auxin-treated and -non-treated *AID-HHF1 hhf2Δ* cells, as compared to WT cells, under heat shock conditions (*Figure 8—figure supplement 2D*). Together, these results suggest that histone H4 plays a role in both *HSP* mRNA multiplexing and the subsequent cell recovery after heat shock.

## mRNAs encoding mitochondrial outer membrane proteins or MAPK proteins also multiplex

To further identify examples of mRNA multiplexing, we employed existing MS2-tagged genes encoding proteins that undergo (or are likely to undergo) complex formation in yeast cells. We first examined whether mRNAs encoding MOMPs multiplex using tagged *UGO1* mRNA. Since the basal expression of *UGO1* mRNA was low (i.e., few puncta observed), we induced expression by growing cells on a non-fermentable carbon source (2% glycerol) before performing RaPID or subsequent fluorescence microscopy experiments. The pulldown of *UGO1* mRNA led to co-precipitation of cohort MOMP mRNAs (e.g., *TOM5, TOM6, TOM22; Figure 9A*), but not *GAPDH* mRNA, suggesting that a specific mRNA multiplex can be detected. Since we initially observed co-precipitation of the mating mRNAs along with *OM45* (*Figure 1A and B*), which codes for another MOMP, we examined for the precipitation of *STE2* mRNA with *UGO1* mRNA. Indeed, *STE2* mRNA could co-precipitate with *UGO1* mRNA, suggesting a possible interaction between MOMP and mating pathway mRNA complexes.

To verify the interaction between MOMP mRNAs, we tagged *TOM6* with the PP7 aptamer in cells expressing MS2-tagged *UGO1* and measured mRNA co-localization using smFISH. We found that 46.3 ± 1.7% (avg. ± SEM) of PP7-tagged *TOM6* mRNA co-localized with MS2-tagged *UGO1* mRNA (*Figure 9B and C*). As control, we visualized native *HSP104* mRNA in co-localization experiments with PP7-tagged *TOM6* mRNA using smFISH. We found 7.7 ± 1.1% of *HSP104* mRNA co-localized with PP7-tagged *TOM6* mRNA (*Figure 9B and C*). We note that less *HSP104* RNA spots were observed on a non-fermentable carbon source (2% glycerol; 6.8 ± 0.4 spots of *HSP104* mRNA; avg. ± SEM), as compared to experiments performed on a glucose-containing medium (e.g., 50.1 ± 0.4 spots of *HSP104* mRNA; avg. ± SEM) (*Figure 8B*), indicating that gene expression was affected. Overall, however, we show that MOMP mRNAs are able to form multiplexes and to co-localize.

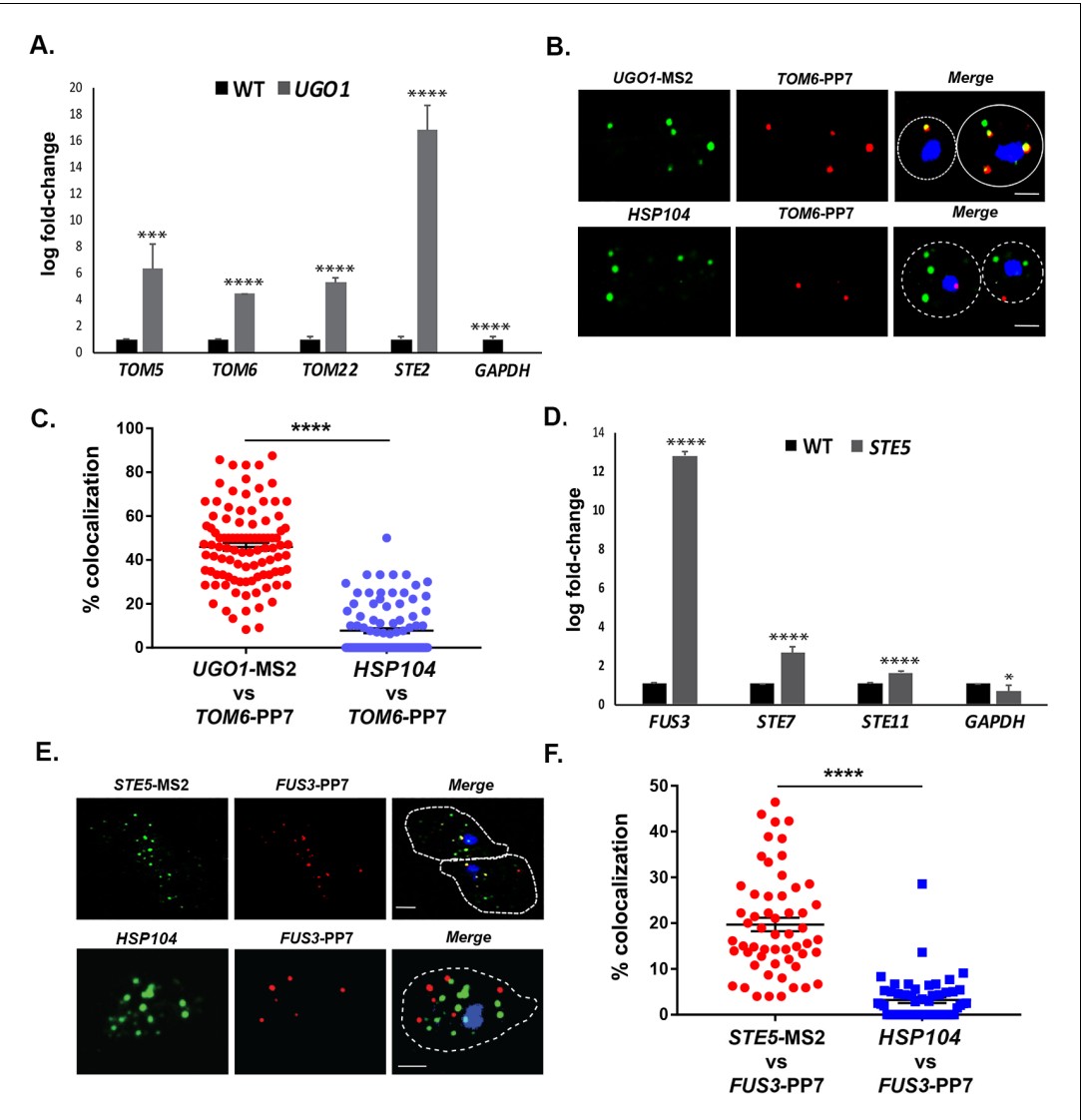

**Figure 9.** mRNAs encoding mitochondrial outer membrane and MAPK proteins multiplex to form different transperons. (**A**) Mitochondrial outer membrane protein (MOMP) mRNAs multiplex. Cells expressing MS2 aptamer-tagged *UGO1* were grown in a medium containing glycerol as a carbon source, prior to fixation and RaPID-qRT-PCR. Three biological replicates were performed and an unpaired t-test was used to compare the *UGO1* pulldown to the wild-type (WT) control pulldown. ****p<0.0001; ***p<0.0005. (**B**) mRNAs encoding MOMP components co-localize. MS2- and PP7 aptamer-tagged *UGO1* and *TOM6* cells, respectively, were grown in a medium containing glycerol as a carbon source and processed for single-molecule fluorescence in situ hybridization (smFISH) labeling with probes complementary to the aptamers or to native *HSP104* mRNA, prior to labeling with DAPI. Dotted lines represent the cell outline observed in brightfield micrographs. *UGO1* – Cy3 labeling; *TOM6* – Cy5 labeling; *HSP104* – Alexa488 labeling; merge – merged Cy3/Alexa-488 and Cy5 windows with DAPI staining. Size bar = 2 µm. The representative image shown is from a single focal plane. (**C**) Scatter plot of data from (B) showing the proportion of co-localized smFISH foci. RNA localization was scored using the FISHquant algorithm (see 'Materials and methods'). Each dot represents a single cell. Black lines indicate avg. ± SEM distribution. (**D**) mRNAs encoding MAPK pathway components multiplex. Cells expressing MS2 aptamer-tagged *STE5* were treated with α-factor (0.5 µM) prior to fixation and RaPID-qRT-PCR. Three biological replicates were performed and an unpaired t-test was used to compare the *STE5* pulldown to the WT control pulldown. ****p<0.0001; *p<0.01. (**E**) mRNAs encoding pheromone-activated MAPK pathway components co-localize. MS2- and PP7 aptamer-tagged *STE5* and *FUS3* cells, respectively, were treated with α-factor (0.5 µM) and processed for smFISH labeling with probes complementary to the aptamers for *STE5* and *FUS3* (MAPK pathway mRNAs) or native *HSP104* mRNA prior to labeling with DAPI. Dotted lines represent the cell outline observed in

*Figure 9 continued on next page*

*Figure 9 continued*

brightfield micrographs. *STE5* – Cy3 labeling; *FUS3* – Cy5 labeling; *HSP104* – Alexa488 labeling; merge – merged Cy3/Alexa-488 and Cy5 windows with DAPI staining. Size bar = 2 µm. The representative image shown is from a single focal plane. (F) Scatter plot of data from (E) showing the proportion of co-localized smFISH foci. RNA localization was scored using the FISHquant algorithm (see 'Materials and methods'). Each dot represents a single cell. Black lines indicate avg. ± SEM distribution.

Finally, we examined whether mRNAs encoding non-secreted cytoplasmic components of the yeast mating pathway (e.g., MAP kinase pathway genes: *STE5*, *FUS3*, *STE7*, and *STE11*) undergo multiplexing like mRNAs encoding their secreted counterparts (*Figures 1* and *2A, B* and *6C*, and *Figure 6—figure supplement 1B*). We tagged *STE5* mRNA with the MS2 aptamer and treated the cells for 1 hr with α-factor (0.5 µM) to induce gene expression. RaPID pulldown of *STE5* mRNA after pheromone induction led to a significant co-precipitation of *FUS3* mRNA, although only slight, but significant, increases in *STE7* or *STE11* mRNA pulldown were detected (*Figure 9D*). We verified the association of MS2-tagged *FUS3* and PP7-tagged *STE5* mRNA in cells using smFISH and observed 19.6 ± 1.4% (avg. ± SEM) co-localization in the presence of α-factor (*Figure 9E and F*). As a control, we visualized *HSP104* mRNA co-localization with PP7-tagged *FUS3* mRNA, but found only 3.2 ± 0.6% co-localization (*Figure 9E and F*). Thus, mRNAs encoding components of the phero-mone-activated MAPK cascade also appear to form specific multiplexes.

## Discussion

Prokaryotes coordinate the expression of genes encoding proteins involved in the same biological process/context by sequential placement in chromosomes in order to generate polycistronic messages (operons) that can be translationally controlled. Perhaps due to the greater complexity observed both at the gene and cellular levels, eukaryotes have preferentially relied on the discontiguous distribution of genes over different chromosomes. In this case, the co-regulation of gene expression presumably becomes dependent on shared transcriptional control elements and specific RBPs that interact selectively with certain transcripts and confer trafficking to sites of co-translation. However, given that the number of RBPs is limited and that they can interact with a wide number of potential transcripts, it is unclear if and how defined subsets of mRNAs are packaged together in order to facilitate co-translation.

We hypothesized that mRNAs encoding proteins involved in the same process might assemble into macromolecular complexes composed of multiplexed transcripts and specific RBPs to form discrete RNP complexes. By performing mRNA pulldown and RNA-seq experiments, we first identified a subset of mRNAs that encode secreted components involved in the mating of α-haploid cells. This mRNP complex included mRNAs encoding the a-pheromone receptor (*STE3*), α-mating factor (*MFα1*, *MFα2*), α-agglutinin (*SAG1*), and a-pheromone blocker (*AFB1*) (*Figures 1A, B* and *2A*, and *Figure 1—figure supplement 1*). Correspondingly, we identified a mating mRNP complex from *MATa* cells that included mRNAs encoding the α-mating factor receptor (*STE2*), a-mating phero-mone (*MFA1*, *MFA2*), and a-agglutinins (*AGA1*, *AGA2*) (*Figure 2B*). Thus, we demonstrated that mRNAs coding for proteins of the same cellular process multiplex into mRNP complexes, presumably to allow for their delivery to the same intracellular site for local translation. Additional experiments in which we identified the existence of other mRNA subsets that multiplex, like those encoding HSPs, MOMPs, and MAPK pathway proteins (*Figures 8* and *9*), clearly support this idea. Thus, mRNA multiplexing in trans appears to form functional transperons (i.e., RNA operons [*Keene, 2007*; *Keene and Tenenbaum, 2002*]) relevant to cell physiology (see below). While we do not know the full extent of RNA multiplexing, we might predict that analogous transperons (i.e., containing other transcripts) form within cells to modulate other physical processes. For example, recent works by Ashe and colleagues have shown that mRNAs encoding either translation factors or glycolytic enzymes co-localize and undergo co-translation in yeast (*Morales-Polanco et al., 2021*; *Pizzinga et al., 2019*). In addition, mRNAs encoding proteins in hetero-oligomeric/multisubunit complexes (proteasome, exocyst, COPI, FAS, ribosome, etc.) might form multiplexes to facilitate polypeptide complex assembly upon co-translation, as shown at the protein level (*Shiber et al., 2018*), even if dedicated assembly chaperones are needed. In our study, we may have missed

multiplexes in the initial RaPID-seq experiment (*Figure 1* and *Figure 1—figure supplement 1*) for several reasons. First, RNA multiplexing may require elevated gene expression, as in the case of the HSP or MOMP complexes (*Figures 8* and *9A–C*), which require induction conditions to visualize the RNA granules, as opposed to the normal growth conditions employed initially. Second, the RaPID-seq pulldowns employed only 0.01% formaldehyde in the crosslinking procedure, as initially published (*Slobodin and Gerst, 2010*), and not the 0.1–0.5% formaldehyde we have employed in the later work (*Zabezhinsky et al., 2016*). Third, the precipitated complexes were stringently washed and, thus, these different factors may have disrupted weaker transperons or missed them entirely.

Importantly, by using a variety of molecular and single-cell imaging techniques (e.g., *Figure 3C*; live fluorescence imaging) we show specific and robust physical interactions between genes located on different chromosomes that encode mRNAs that form transperons (*Figure 7* and *Figure 7—figure supplement 1A and B*). This observation of allelic coupling is strengthened by the work of *Chowdhary et al., 2019*, where the intergenic association of yeast HSP genes was shown. Other chromatin conformation capture studies have revealed intergenic associations that lead to the so-called transcription factories associated with different transcriptional regulators (e.g., promoters, transcription factors). For example, *Schoenfelder et al., 2010* showed interchromosomal interactions between Klf4-regulated globin genes in fetal liver cells that allowed for co-regulated gene transcription. *Papantonis et al., 2012* revealed that TNF-α responsive coding and miRNA genes in human endothelial cells undergo intrachromosomal interactions regulated by NFκB upon cytokine induction. Thus, specialized transcriptional factories created through allelic coupling may be a common mechanism for co-regulated gene expression in eukaryotes.

How allelic coupling comes about to form these factories and resulting transperons is not yet clear. However, we identified a role for histone H4 in the assembly of the mating transperon (*Figures 3C, E* and *4C, D*, and *Figure 4—figure supplement 1E*) and its physiological consequences (i.e., pheromone responsiveness of both mating partners and mating; *Figure 4A and B*, and *Figure 6—figure supplement 2A*). RaPID-MS pulldowns of mating mRNAs encoding secreted proteins identified the H4 paralog, Hhf1, as binding to the *STE2*, *MFA1*, and *MFA2* mRNAs, but not to *ASH1* mRNA or *STE2* mRNA lacking its 3′UTR (*Figure 3—figure supplement 1*). Thus, specific H4-RNA interactions are likely to be involved in transperon assembly. Hhf1 pulldowns directly precipitated the mating transperon (*Figure 3C and E*) and the deletion of either paralog (*HHF1* or *HHF2*) led to defects in RNA multiplexing (*Figure 4C and D*) and mating (*Figure 4A and B*). Likewise, a combination of *HHF2* deletion and an auxin-induced degradative form of Hhf1 resulted in a complete block in mating and loss of mRNA co-localization under conditions in which cell viability was not affected (*Figure 4B* and *Figure 4—figure supplement 1C and E*). Similar results were obtained using the HSP mRNAs, which also form physiologically functional transperons and co-localize in a manner dependent upon histone H4 (*Figure 8*, *Figure 8—figure supplement 1D*, and *Figure 8—figure supplement 2*). This suggests that both Hhf1 and Hhf2 are important for the formation of the mRNA complexes identified here.

Mutations in the amino terminal histone H4 acetylation sites had similar effects as gene deletions, whereby inactivating K-to-R mutations inhibited mating (*Figure 5B and C*). In contrast, K-to-Q mutations (or H4 overexpression) improved mating (*Figure 5B*). Thus, histone H4 paralogs, but no other histones (*Figure 4A* and *Figure 8—figure supplement 1C*), appear to affect the transmission of physiological signals relevant to RNA multiplexing. More work is required to demonstrate the role histone H4 plays in transperon formation and whether H4 is present in these multiplexes post-assembly and export from the nucleus. Although we found that fluorescent protein-tagged Hhf1 labeled the nucleus, we were unable to observe H4 in the cytoplasm or show co-localization there with *STE2* mRNA (data not shown). Thus, we do not know whether this non-canonical interaction of H4 with mRNA is exclusive to the nucleus or not. Canonical histone functions relate mainly to DNA-protein interactions that facilitate nucleosome formation. However, the requirement of acetylation for H4 function in mating (*Figure 5A–C*) suggests that histone H4 might participate in the formation of a DNA-RNA intermediate upon gene transcription. This mechanism potentially allows for RNA multiplexing, provided that the loci encoding mRNAs intended for assembly in trans also are in close apposition, as evidenced here (*Figure 7B and C*, and *Figure 7—figure supplement 1A,B and F*). Results showing that a transcription inhibitor affects allelic coupling imply that gene apposition and transcription may be connected (*Figure 7—figure supplement 1E*). Moreover, the fact that H4

deletion also affects allelic coupling may implicate an involvement of chromatin architecture in transperon formation (*Figure 7—figure supplement 1F*).

Our lab is interested in how mRNA localization affects cell physiology, and a previous work has demonstrated that yeast mSMPs (including *MFA2*) localize to ER in the cell body (*Kraut-Cohen et al., 2013*). Here, we verified that endogenously expressed *STE2* and *AGA1* mRNAs also localize to ER in the cell body, both before and after treatment with pheromone (*Figure 2C*). Moreover, these mRNAs co-localize both prior to and post nuclear export (*Figure 2D–F*). This finding demonstrates once again that mSMPs are not asymmetrically localized to the polarized extensions of yeast (e.g., bud and shmoo tips), unlike *ASH1* and polarity-establishing mRNAs (e.g., *SRO7, SEC4, CDC42*) in budding cells, which is mediated by She2 (*Aronov et al., 2007*), or *SRO7* and *FUS3* mRNAs in shmooing cells, which is mediated by Scp160 (*Gelin-Licht et al., 2012*). It also demonstrates that RNA multiplexing and transperon formation occur within the nucleus (*Figures 2D–F* and *8B*). Since this work and previous studies have revealed that mRNAs encoding secreted mating pathway components, MOMPs, and pheromone-activated MAPK pathway components localize to the ER, mitochondria, and ER of the shmoo tip, respectively (*Gadir et al., 2011*; *Gelin-Licht et al., 2012*; *Zabezhinsky et al., 2016*), we predict that these mRNAs are targeted as intact transperons. Moreover, as Scp160 binds MAPK pathway mRNAs (e.g., *FUS3, STE7, KAR3*) and delivers them to the shmoo tip upon pheromone treatment (*Gelin-Licht et al., 2012*), we predict that Scp160 might traffick this MAPK transperon.

An earlier work has (*Aronov et al., 2015*) suggested that mRNAs encoding mating components, like *MFA2*, localize preferentially to P-bodies in the shmoos and shmoo tips of pheromone-treated yeast, and that P-bodies are necessary for transmission of the mating signal and subsequent mating. However, these results were obtained under conditions of mRNA overexpression. In contrast, we found that neither endogenously expressed MS2-tagged *MFA2* (*Haimovich et al., 2016*) nor *STE2* mRNA (*Figure 6—figure supplement 2C*) localize to P-bodies, nor do P-bodies form under normal growth conditions conducive to the expression of these mRNAs and to mating (*Figure 6—figure supplement 2C*; top panel). In addition, we show here that *MFA2* and other mating mRNAs (e.g. *MFA1, STE2*, and *AGA1*) do not localize to the shmoos or shmoo tips, but rather to ER present in the cell body (*Figure 2C* and *Figure 2—figure supplement 1*). Moreover, the deletion of genes necessary for P-body formation (e.g., *PAT1, DHH1*) had no significant effect upon mating (*Figure 6—figure supplement 2B*). As mating pathway mRNAs localize to P-bodies only upon stress conditions (e.g., gene overexpression, glucose starvation) (*Haimovich et al., 2016* and *Figure 6—figure supplement 2C*), it would appear that P-bodies are not sites for mating mRNA localization or transmission of the mating signal under normal growth conditions. Our findings re-emphasize the importance of (i) measuring mRNA localization under native conditions of gene expression; (ii) considering the possibility that RNA overexpression and/or the insertion of RNA aptamers may induce artifacts (e.g., mRNA mislocalization, P-body formation, or both); and (iii) examining mRNA integrity/localization using additional techniques (e.g., Northern analysis, RNA-seq, smFISH) (*Garcia and Parker, 2015*; *Haimovich et al., 2016*; *Heinrich et al., 2017*) for secondary validation.

As *cis*-acting RNA elements may work in conjunction with specific RNA-binding proteins to confer multiplexing, we analyzed the coding regions and 3'UTRs of the mating mRNAs from the two yeast haplotypes. Sequence analysis revealed the presence of conserved motifs present in the mating mRNAs from either *MATa* or *MATα* cells (*Figure 6A*). We mutated the *MATa* sequence element in the *AGA2* gene and found that it affected transperon assembly similar to the deletion of *AGA2* (*aga2Δ*) (*Figure 6B*). Importantly, this correlated with an inhibition in cellular responsiveness (i.e., shmooing) toward its mating partner, as well as the responsiveness of its mating partner (*Figure 6—figure supplement 2A*), resulting in significant defects in mating (*Figure 6C*). Thus, defects in transperon formation result in significant changes in cell physiology, and we predict that a yet unidentified RBP interacts with this element to facilitate RNA assembly and, possibly, trafficking. The *MATa* element is unlikely to be the H4-interacting motif itself, since it was not identified among the mating mRNAs from *MATα* cells, which have other shared motifs (*Figure 6—figure supplement 1C*). Likewise, removal of the 3'UTR from *STE2* appeared to reduce H4 binding (*Figure 3—figure supplement 1*), possibly indicating its role in histone association. Additional RaPID-MS experiments should help reveal the identity of this protein, as well as those interacting with the motifs in mating mRNAs from *MATα* cells (*Figure 6—figure supplement 1C*), and genetic analyses should reveal their contribution to the mating process.

Finally, we note that in a few cases, transcripts from genes that neighbor the target gene were enriched in MS2-tagged target mRNA pulldowns in our RaPID-seq experiment (*Figure 1A* and *Figure 1—figure supplement 1*; see *Supplementary file 1* for RNA-seq data), as compared to the bead control. For example, we observed that *PET122* (YER153C) mRNA precipitated with *OXA1* (YER154W); *SPE1* (YKL184W) with *ASH1* (YKL185W); *SNC1* (YAL030W) with *MYO4* (YAL029C); *SNC2* (YOR327C) with *MYO2* (YOR326W); and *ALY2* (YJL084C) with *EXO70* (YJL085W). It is possible that these genes are co-expressed and that their chromatin-associated RNA transcripts precipitate during RaPID, prior to DNase treatment. We did not examine these interactions further. Nonetheless, this possibility does not negate our hypothesis supporting the existence of RNA multiplexes/transperons, which was validated by a variety of means (e.g., RaPID-qRT-PCR, smFISH, and physiological assays).

To the best of our knowledge, the results presented here support the concept that eukaryotic mRNP complexes/transperons are the functional equivalents of bacterial operons, as first proposed by *Keene, 2007* and *Keene and Tenenbaum, 2002*. Thus, eukaryotes may circumvent polycistronicity by assembling relevant mRNAs into functional complexes that confer co-translation. This is likely to occur via physical intergenic interactions that couple individual alleles concurrent with transcription to form transperons that can undergo export and translation at different sites within the cell (*Figure 10A*). A schematic describing mRNA assembly into functional transperons encoding secreted mating pathway components via histone H4 mediation is shown in *Figure 10B*. While this work implies that there are higher order principles governing the selection of mRNAs for mRNP complex assembly, further study is required to uncover the entire mechanism by which mRNAs are recruited and assembled together into multiplexes, and to elucidate how histones and other proteins link mRNAs to fully formed transperons.

## Materials and methods

### Yeast strains, genomic manipulations, growth conditions, and plasmids

Yeast strains used are listed in *Supplementary file 2*. Standard LiOAc-based protocols were employed for the introduction of plasmids and PCR products into yeast. For RNA and protein localization experiments, cells were inoculated and either grown at 26°C to mid-log phase (O.D.$_{600}$=~1) in a standard rich growth medium containing 2% glucose (YPD) or in a synthetic medium containing 2% glucose (e.g., synthetic complete [SC]) and selective SC drop-out medium lacking an amino acid. Induction of the methionine starvation-inducible plasmids (e.g., pUG36-based MS2-CP-GFP(x3)) was performed by transferring cells grown to mid-log phase to a synthetic medium lacking methionine and subsequent growth of the cells for 1 hr with shaking at 26°C. For growth tests, $2 \times 10^7$ yeast cells grown to mid-log phase and normalized for absorbance at OD$_{600}$ were serially diluted 1:10, plated by drops onto a solid synthetic selective medium, and grown at different times/temperatures before photo-documentation. Three biological replicas were performed for each growth test.

Plasmids used are listed in *Supplementary file 3*. Gene deletions were made using the PFA6-based plasmids (*Longtine et al., 1998*) and PCR amplification of the deletion cassettes. Point mutations at genomic loci were made using a CRISPR/Cas9-based system (*Mans et al., 2015*) that was modified to express both Cas9 and gRNA from the same pRS426 plasmid. gDNAs were designed by CHOPCHOP software (*Labun et al., 2016*) and introduced between *SNR52* promoter and gRNA scaffold using the FastCloning technique (*Li et al., 2011*). The repair long primer was designed to have homologous overlapping sequences of 45 bp before and after the site of mutation, and was introduced into yeast along with Cas9 and gDNA in one transformation.

A *MAT*a W303 strain containing *GFP-lacI* at the *HIS3* locus and *tetR-3xCFP* at the *ADE1* locus (*Dovrat et al., 2018*) was modified by replacing *tetR-3xCFP* with *tetR-tdTomato*. The integration of *lacO* repeats and *tetR* repeats was performed as described in *Dovrat et al., 2018*. A selection marker (*NAT*) was amplified with flanking sequences homologous downstream of the target gene (e. g., *STE2*, *AGA2*, *ASH1*) in the genome and transformed into yeast. After validation of correct insertion by PCR, a linearized plasmid containing either the *lacO* or *tetR* array (*Dovrat et al., 2018*), a selection marker (e.g., *LEU2* or *TRP1*), and flanking *NAT* gene target sequences was transformed. This leads to replacement of the *NAT* selection marker by the linearized array. The integration of

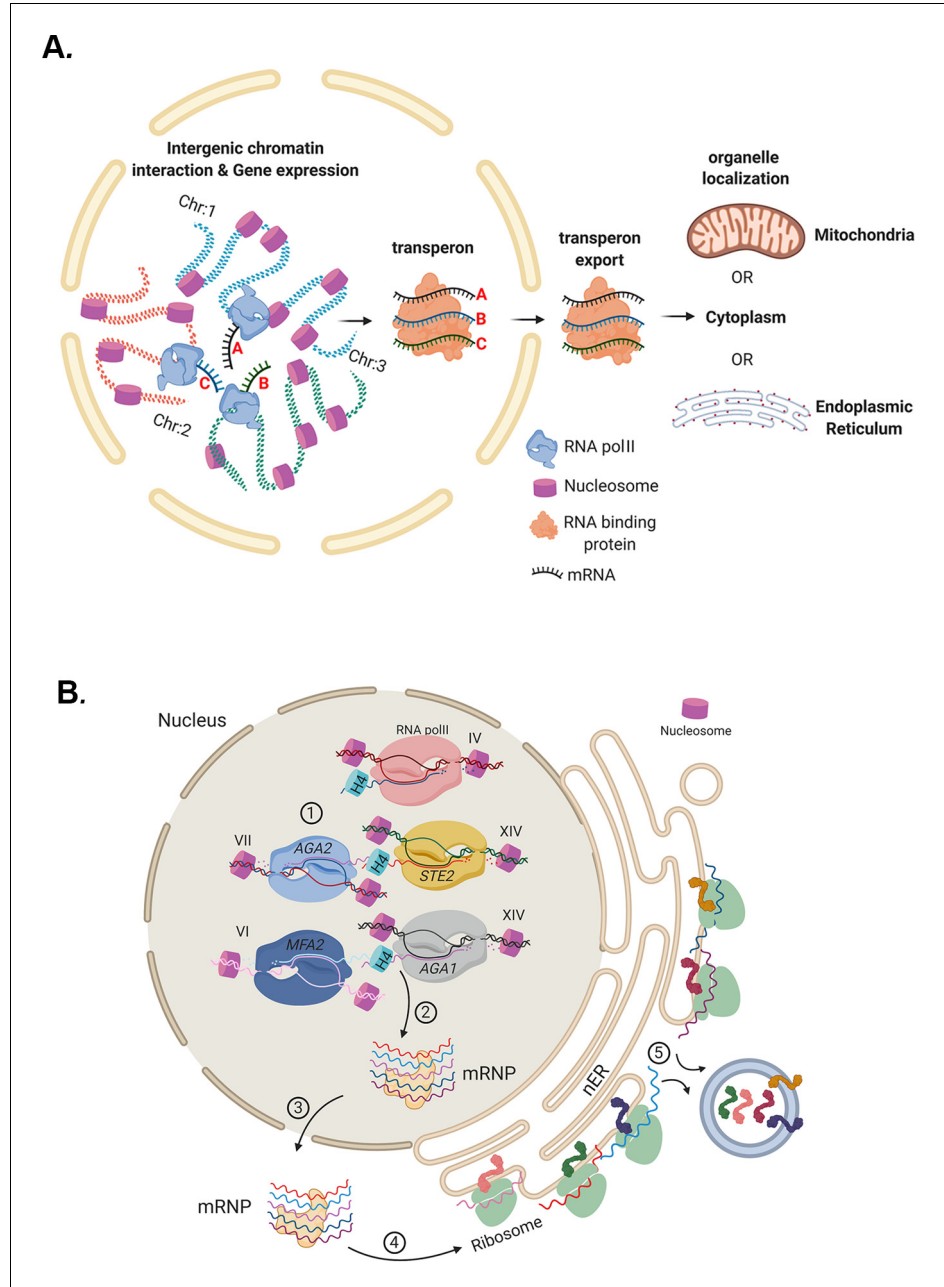

**Figure 10.** Models for mRNP transperon formation and its role in the co-translation of proteins involved in cell physiology. (**A**) A schematic illustrating the mechanism of mRNA multiplexing and its role in cell physiology. Gene loci located on different chromosomes undergo allelic coupling under different growth conditions to form intergenic associations. Upon transcription, intergenic association allows for the mRNAs to assemble into a mRNA ribonucleoprotein (mRNP) complex (transperon) that is exported to the cytoplasm and confers mRNA co-localization at different organelles (mitochondria, ER, cytoplasm). RNA co-localization and co-translation allow for coordination of the pathways responsible for the control of cell physiology. (**B**) mRNA multiplexing and transperon formation control the mating pathway in yeast. Yeast (*MAT*a cells, as shown) express mRNAs encoding soluble secreted and membrane proteins (e.g., Mfa1, Mfa2, Aga1, Aga2, and Ste2, respectively) from genomic loci located on different chromosomes (as listed) that undergo allelic coupling to form intergenic associations. Histone H4 paralogs (Hhf1 and -2) are required for the mRNAs to assemble into a mating mRNA transperon that is exported to the cytoplasm and confers RNA co-localization at the endoplasmic reticulum peripheral to the nucleus (nER). Co-translational translocation of these proteins at the ER may coordinate the pathways responsible for the cellular responsiveness to pheromone of the opposite mating type (i.e., via both Ste2 signaling and Aga1 and Aga2

*Figure 10 continued on next page*

function in cell adhesion) and transmission of the cell's own mating signal to cells of the opposite mating type (i.e., via Mfa1 and Mfa2 secretion).

each array (e.g., *lacO* or *tetR*) into the target genes was performed sequentially and after visual confirmation of the first integration using fluorescence microscopy.

Growth assays to determine the doubling time of WT (BY4741) and histone mutants after heat shock involved growing the cells to mid-log phase at 30°C to O.D.$_{600}$ = 0.4–0.8. A portion of culture (0.5 O.D.$_{600}$ units) was maintained at 30°C and the remainder (0.5 O.D.$_{600}$ units) was subjected to heat shock at 50°C for 60 min (*Figure 8—figure supplement 1C and D*). Cells were then pelleted and resuspended in 1 ml rich media (YPD; 24°C). A volume of 60 µl of cells was transferred to 120 µl of YPD media in 96-well plates and the O.D.$_{600}$ values were measured at 30°C in 30 min intervals for 60 cycles using a Tecan liquid handling robot. For experiments involving heat shock and the auxin-mediated depletion of Hhf1 (*Figure 8—figure supplement 1D*), cells were grown to mid-log phase and auxin (3-IAA; 4 mM final concentration) was added to the medium for 3.5 hr at 30°C. After this, cells were exposed to heat shock at 50°C for 15, 30, or 60 min and then pelleted and resuspended in 1 ml YPD and diluted and grown, as described above.

## Fluorescence microscopy

Cells grown for fluorescence microscopy were collected and fixed in a solution containing 4% (w/v) paraformaldehyde and 4% (w/v) sucrose for 20 min at room temperature (RT), washed once, resuspended in 0.1 M KPO4/1.2 M sucrose (w/v) buffer solution, and kept at 4°C. For staining of mitochondria, Mitotracker Red (Molecular Probes, Eugene, OR) was added to mid-log phase-grown cells to a final concentration of 0.2 µM for 20 min before harvesting. After collection, the cells were pelleted, washed once in fresh medium at the appropriate temperature, and analyzed by fluorescence microscopy. Representative images were acquired using either a Zeiss LSM780 or LSM710 confocal and a Plan-Apochromat x100/1.40 oil objective (Carl Zeiss AG, Oberkochen, Germany). The following wavelengths were used (using separate tracks): LSM510 – GFP, excitation at 488 nm/emission at >505 nm; red fluorescence, excitation at 561 nm/emission at >575 nm; LSM710 – GFP, excitation at 480 nm/emission at 530 nm; monomeric RFP, excitation at 545 nm/emission at 560–580 nm. For statistical analyses, at least three identical individual experiments were carried out. Unless otherwise stated, a total of 100 RNA granules per experiment were scored for the mRNA localization in cells, while 100–150 cells per experiment were scored for protein localization. For the scoring of either RNA or protein localization, the average of n = 3 experiments (± standard error of the mean; SEM) is given. For p-value calculations, a Student's t-test (two-tailed, paired) was done.

## Single-molecule FISH

Yeast expressing MS2-tagged and PP7-tagged genes or non-tagged BY4741 WT cells were grown to mid-log phase prior to fixation. Cells were pelleted and suspended in phosphate-buffered saline (PBS), fixed in the same medium upon the addition of paraformaldehyde (4% final concentration), and incubated at RT for 45 min with rotation. Cells were washed three times with ice-cold Buffer B (0.1 M potassium phosphate buffer, pH 7.5, containing 1.2 M sorbitol), after which cells were spheroplasted in 1 ml of freshly prepared spheroplast buffer (Buffer B supplemented with 20 mM ribonucleoside vanadyl complexes [Sigma-Aldrich, St. Louis, MO], 20 mM β-mercaptoethanol, and lyticase [Sigma-Aldrich, St. Louis, MO] [25 U per O.D.$_{600}$ unit of cells]) for 10 min at 30°C. The spheroplasts were centrifuged for 5 min at a low speed (2500 rpm; 660 x*g*) at 4°C and washed twice in ice-cold Buffer B. Spheroplasts were then resuspended in 750 ul of Buffer B, and approximately 2.5 O.D.$_{600}$ units of cells were placed on poly-L-lysine-coated coverslips in 12-well plates and incubated on ice for 30 min. Cells were carefully washed once with Buffer B, then incubated with 70% ethanol for several hours to overnight at −20°C. Afterwards, cells were washed once with SSCx2 (0.3 M sodium chloride, 30 mM sodium citrate), followed by incubating with Wash buffer (SSCx2 with 10% formamide), for 15 min at RT (~23°C). Next, 45 µl of hybridization buffer (SSCx2, 10% dextran sulfate, 10% formamide, 2 mM ribonucleoside vanadyl complexes, 1 mg/ml *E. coli* tRNA, and 0.2 mg/ml bovine serum albumin [BSA]) containing 50 ng/µl MS2-Cy3 and PP7-Cy5 probe mix was

placed on parafilm in a hybridization chamber. Coverslips with the immobilized cells were placed face down on top of the hybridization buffer and were incubated overnight at 37°C in the dark. After probe hybridization, cells were incubated twice in Wash buffer for 15 min at 37°C. Next, the cells were washed once with SSCx2 containing 0.1% Triton X-100, incubated with SSCx2 supplemented with 0.5 μg/ml DAPI for 1 min at RT, and finally washed with SSCx2 for 5 min at RT. Cells were mounted with Prolong Glass (Thermo Scientific) mounting media on clean microscope slides. Samples were imaged using PCO-Edge sCMOS camera controlled by VisView installed on a VisiScope Confocal Cell Explorer system (Yokogawa spinning disk scanning unit); CSU-W1 and an inverted Olympus microscope (×100 oil objective; excitation wavelengths: GFP – 488 nm; Cy3 – 555 nm; Cy5 – 640 nm). More than 150 cells with mRNA spots were scored for each sample using the FISH-quant program (*Mueller et al., 2013*) (https://bitbucket.org/muellerflorian/fish_quant) to analyze images of single cells. Co-localization between the two channels was calculated as a linear assignment problem solved with the Hungarian algorithm. We used Matlab functions hungarianlinker[2] and munkres[3] developed by *Mueller et al., 2013* for this purpose. We also crosschecked co-localization (defined by overlap between the signals) manually for verification. Greater than 150 cells were scored in each experiment and the average ± SEM is given.

In addition to using fluorescent MS2 and PP7 probes, we analyzed the co-localization of endogenous genes in some experiments (e.g., *STE2-AGA1, HSP104-SSA2*) using gene-specific 20 nt antisense oligonucleotides bearing a 59 nt Flap sequence [Flap-Z, Flap-X] and to which complementary fluorescent-labeled oligonucleotides (488-Alexa, Cy3) were annealed (*Pizzinga et al., 2019*; *Tsanov et al., 2016*).

## RaPID procedure for precipitation of RNP complexes and RNA sequencing

The pulldown of MS2 aptamer-tagged mRNAs and detection for bound RNAs and proteins was performed using the RaPID procedure, essentially as described (*Slobodin and Gerst, 2010*; *Slobodin and Gerst, 2011*). Yeast strains bearing endogenously expressed genes tagged with the MS2 aptamer (*Haim et al., 2007*) were grown in a volume of 400 ml to mid-log phase at 26°C with constant shaking to an O.D.$_{600}$=~0.8. Cells were centrifuged in a Sorvall SLA3000 rotor at 1100 x$g$ for 5 min, resuspended in 200 ml of complete synthetic medium lacking methionine in order to induce the expression of the MS2-CP-GFP-SBP protein, and grown for an additional 1 hr. The cells were collected by centrifugation as described above, washed with PBS buffer (lacking Ca$^{++}$ and Mg$^{++}$), and transferred into a 50 ml tube and pelleted as above. For the experiment shown in *Figure 1*, proteins were crosslinked by the addition of 8 ml PBS containing 0.01% formaldehyde and incubated at 24°C for 10 min with slow shaking. For all other experiments, 0.05% formaldehyde was used. The crosslinking reaction was terminated by adding 1 M glycine buffer, pH = 8.0, to a final concentration of 0.125 M with additional shaking for 2 min. The cells were then washed once with ice-cold PBS buffer, and the pellet was flash frozen in liquid nitrogen and stored at −80°C.

For cell lysis and RNA pulldown, cell pellets were thawed upon the addition of ice-cold lysis buffer (20 mM Tris-HCl at pH 7.5, 150 mM NaCl, 1.8 mM MgCl$_2$, and 0.5% NP40 supplemented with aprotinin [10 mg/ml], phenylmethylsulfonyl fluoride (PMSF) [1 mM], pepstatin A [10 mg/ml], leupeptin [10 mg/ml], 1 mM dithiothreitol (DTT), and 80 U/ml RNAsin [Promega]) at 1 ml per 100 O.D.$_{600}$ U, and 0.5 ml aliquots were then transferred to separate microcentrifuge tubes containing an equal volume of 0.5 mm glass beads, and vortexed in an Vortex Genie Cell Disruptor (Scientific Instruments, New York) shaker at maximum speed for 10 min at 4°C. Glass beads and unbroken cells were sedimented at 4°C by centrifugation at 1700 x$g$ for 1 min, and the supernatant was removed to new microcentrifuge tubes and centrifuged at 11,000 x$g$ at 4°C for 10 min. The total cell lysate (TCL) was then removed to a fresh tube and protein concentration was determined using the microBCA protein determination kit (Pierce). Protein (10 mg of total cell lysate) was taken per pulldown reaction. In order to block endogenous biotinylated moieties, protein aliquots were incubated with 10 μg of free avidin (Sigma) per 1 mg of input protein for 1 hr at 4°C with constant rotation. In parallel, streptavidin-conjugated beads (Streptavidin-sepharose high performance, GE Healthcare) were aliquoted to microcentrifuge tubes according to 5 μl of slurry per 1 mg of protein (not >30 μl overall), washed twice with 1 ml of PBS, once with 1 ml of lysis buffer, and blocked with a 1:1 mixture of 1 ml of lysis buffer containing yeast tRNA (Sigma; 0.1 mg/100 ml of beads) and 1 ml of 4% BSA in PBS at 4°C for 1 hr with constant rotation. Following blocking, beads were washed twice in 1 ml of lysis buffer.

Pulldown was performed by adding the indicated amount of avidin-blocked TCL to the beads, followed by incubation at 4°C for 2–15 hr with constant rotation. Yeast tRNA was added to the pulldown reaction (0.1 mg/tube) to reduce nonspecific interactions. We used standard 1.7 ml microcentrifuge tubes when working with small volumes of TCL or 15 ml sterile polypropylene centrifuge tubes with larger volumes. Following pulldown, the beads were centrifuged at 660 x$g$ at 4°C for 2 min, the supernatant then removed, and the beads washed three times with lysis buffer (e.g., 1 ml volume washes for small tubes, 2 ml for large tubes), twice with wash buffer (20 mM Tris, pH 7.5, 300 mM NaCl, and 0.5% Triton X-100), all performed at 4°C (each step lasting 10 min with rotation). The beads were then equilibrated by a final wash in 1–2 ml of cold PBS, pelleted by centrifugation as above, and excess buffer aspirated. For elution of the crosslinked RNP complexes from the beads, 150 ml of PBS containing 6 mM free biotin (Sigma) was added to the beads, followed by 1 hr of incubation at 4°C with rotation. After centrifugation at 660 x$g$ for 2 min, the eluate was transferred into a fresh microcentrifuge tube, re-centrifuged, and transferred into another tube to assure that no beads were carried over. To reverse the crosslink, the eluate was incubated at 70°C for 1–2 hr with an equal volume of 2X crosslink reversal buffer (100 mM Tris, pH7.0, 10 mM EDTA, 20 mM DTT, and 2% sodium dodecyl sulfate [SDS]) for RNA analysis or with an appropriate volume of 5X protein sample buffer (5X: 0.4 M Tris at pH 6.8, 50% glycerol, 10% SDS, 0.5 M DTT, and 0.25% bromophenol blue) for protein analysis using SDS-PAGE. For RaPID pulldowns and RT-PCR analysis, three biological replicas were performed each and the average ± standard deviation is given. For p-value calculation, a Student's t-test (two-tailed, paired) was performed.

For the RNA sequencing and analysis shown in *Figure 1* and *Figure 1—figure supplement 1*, libraries were constructed for RNA-Seq from the affinity-purified RNA, as described for the control (non-strand-specific) library in *Levin et al., 2010*, except that the oligo(dT) selection and RNA fragmentation procedures were omitted. In addition, only random hexamers were used for cDNA synthesis and Ampure beads (Beckman-Coulter, IN) were used for size selection. Libraries were sequenced using an Illumina HiSeq2000 sequencer (paired-end 76 base reads) to a depth of ~$10^7$ reads. Gene expression levels were estimated from the RNA-Seq data using Kallisto (*Bray et al., 2016*) and targeting the *Saccharomyces cerevisiae* (*S. cerevisiae*) reference transcriptome (derived from the Saccharomyces Genome Database and leveraging the *Saccharomyces cerevisiae* S288C genome version R64-2-1), where we added 100 bases to the 3' UTR of each coding region. Differentially expressed genes were identified using edgeR (*Robinson et al., 2010*), with the dispersion parameter manually set to 0.1. Those genes reported as at least fourfold differentially expressed (FDR <0.001) were retained as significantly differentially expressed. After log transforming the data (log2(TPM +1)), the gene expression values from the control sample were subtracted from experimental samples and plotted in a heat map illustrating expression levels as compared to the control sample. Expression quantitation, differential expression, and plotting of heat maps were facilitated through use of the transcriptome analysis modules integrated into the Trinity software suite (*Haas et al., 2013*).

For protein analysis, samples of the eluate were electrophoresed on 10% SDS-PAGE gels, followed by silver staining using the Pierce Silver Stain kit (ThermoFisher) to identify bands of interest. Bands were excised from gels and processed for reversed-phase nano-liquid chromatography-electrospray ionization tandem mass spectrometry at the Israel National Center for Personalized Medicine (Weizmann Institute, Rehovot, Israel).

## Immunoprecipitation

IP of protein-RNA complexes was performed as described in *Gelin-Licht et al., 2012*. Briefly, WT (BY4741) cells, WT cells expressing HA-tagged Hhf1, and WT cells expressing both MS2 aptamer-tagged *STE2* and HA-Hhf1 were grown to mid-log phase (O.D.$_{600}$=~0.5) in 200 ml synthetic selective medium at 26°C. Cultures were centrifuged at 3000 x$g$ for 5 min at 4°C and resuspended in 1 ml of lysis buffer (20 mM Tris-Cl at pH 7.5, 150 mM KCl, 5 mM MgCl$_2$, 100 U/ml RNasin, 0.1% NP-40, 1 mM DTT, 2 μg/ml aprotinin, 1 μg/ml pepstatin, 0.5 μg/ml leupeptin, and 0.01 μg/ml benzamidine). Cells (100 O.D.$_{600}$ units each) were broken by vortexing with glass beads (0.5 mm size) for 10 min at 4°C and centrifuged at 10,000 ×$g$ for 10 min at 4°C to yield the TCL. For IP, 3 mg of TCL was diluted in lysis buffer (final volume, 1 ml) and subjected to IP with 3 μl of monoclonal anti-HA antibody (16B12; Covance) overnight at 4°C with rotation. Next, 50 μl of a protein G agarose slurry (Santa Cruz Biotechnology, Dallas, TX) was added and the samples were incubated for an additional 1 hr at

4°C with rotation. The precipitates were washed (×5) with lysis buffer and protein–RNA complexes were eluted by incubation at 65°C for 30 min in 300 µl of elution buffer (50 mM Tris-HCl at pH 8.0, 100 mM NaCl, 10 mM EDTA, 0.1% SDS, and 100 µg/ml proteinase K in RNase-free water).

For RNA detection, RNA was extracted from each sample using the Epicentre MasterPure kit (Lucigen, WI), treated with DNase (2 hr), and 15.1 µl (from the total of 35 µl) was taken for reverse transcription (RT). For RT-PCR, 1 µl of the RT reaction was diluted (x4) and 1 µl was taken for PCR detection using specific primers for *STE2*, *MFA1*, *MFA2*, as well as *EXO70*, *ASH1*, *SRO7*, and *UBC6*, as potential controls. For qRT-PCR, 1 µl of the RT reaction was diluted (x4) and 1 µl taken for qRT-PCR detection using SYBR Green (as described above) with specific primers for *AGA1*, *AGA2*, *STE2*, *MFA1*, *MFA2*, and *UBC6*.

For Western blot analysis, samples (30 µg) were electrophoresed on 15% SDS-PAGE gels. Protein detection using anti-HA (Covance 16B12, BioLegend, Dedham, MA) or anti-actin antibodies (Cell Signaling, Danvers, MA) was performed using the Amersham ECL Western Blotting Detection Kit (GE Healthcare Life Sciences). Quantification of protein bands in gels was performed using Gel-Quant.NET software provided by biochemlabsolutions.com.

## Quantitative mating assay

Quantitative analysis of yeast mating was performed based on the complementation of auxotrophic markers present in the *MAT*a and *MAT*α cells after crossing (i.e., the *met15* mutation in BY4741 *MAT*a cells and *lys2* mutation in BY4742 *MAT*α cells are complemented upon mating to yield diploids that grow on a medium lacking methionine and lysine). Cells from each mating type were grown to mid-log phase at 26°C to O.D.$_{600}$ = 0.3–0.4. $10^6$ cells from each mating type (in 150 µl volume), as well as 1:1 *MAT*a and *MAT*α mixtures, were collected by filtration on individual cellulose nitrate filters (25 mm diameter, 0.45 µm pores; BA-85 Schleicher and Schuell, Maidstone, UK) placed on the top of a working filtration surface of a disposable 500 ml filtration apparatus (Corning, NY). After filtration to remove the liquid medium, each filter was placed cell-side up on YPD plates for 3 hr at 30°C, prior to resuspension in 10 ml of SC-Met,-Lys in a sterile 50 ml tube, followed by vortexing, serial dilution (1:10,100,1000), and plating of 100 µl of the dilutions onto SC-Met,-Lys plates to select for diploids. Mating efficiency was determined by colony scoring after 3 days at 26°C. Three biological replicas were performed for each experiment and the average ± standard deviation is given. For p-value calculation, a Student's t-test (two-tailed, paired) was performed.

## Quantitative yeast pheromone responsiveness (shmoo formation)

To determine the responsiveness of cells toward their mating partner (e.g., WT *MAT*α cells toward mutant *MAT*a cells and vice versa), the level of shmoo formation in each strain was measured. Cells from each mating type were grown to mid-log phase at 26°C to O.D.$_{600}$ = 1–2. Next, $10^6$ cells from each mating type were mixed (1:1) in a final volume of 150 µl and collected by filtration on individual cellulose nitrate filters (25 mm diameter, 0.45 µm pores; BA-85 Schleicher and Schuell, Maidstone, UK) placed on the working filtration surface of a disposable 500 ml filtration apparatus (Corning, NY). After filtration to remove the liquid medium, each filter was placed cell-side up on YPD plates for 3 hr at 30°C, prior to resuspension in 1 ml of 1X TE, followed by vortexing and pelleting of the cells. Cells were then fixed for 15 min by resuspension in 4% (w/v) paraformaldehyde and 4% (w/v) sucrose for 20 min at RT, washed once, suspended in 0.1 M KPO4/1.2 M sucrose (w/v) buffer solution, and kept at 4°C. Images were acquired using Zeiss Plan-Apochromat x100/1.40 oil objective (Carl Zeiss AG, Oberkochen, Germany). For statistical analyses, at least three identical individual experiments were carried out. Unless otherwise stated, a total of 100 or more cells per strain per experiment were scored for the formation of shmoo projections. For scoring of shmooing efficiency, the average of n = 3 experiments is given. For p-value calculation, a one-way ANOVA with multiple comparisons was performed.

## Chromosome conformation capture

The quantitative chromosome conformation capture method, TaqI-3C (*Chowdhary et al., 2019*; *Hagège et al., 2007*), was employed with modifications for use in yeast. BY4741 *MAT*a cells were cultured to 0.6–0.8 O.D.$_{600}$ in a volume of 50 ml to early log phase at 30°C and then crosslinked with 1% formaldehyde. Crosslinked cells were then subjected to glass bead-mediated lysis in FA lysis

buffer (50 mM HEPES at pH 7.9, 140 mM NaCl, 1% Triton X-100, 0.1% sodium deoxycholate, 1 mM EDTA, 1 mM PMSF) for two cycles (20 min each) of vortexing at 4°C. Cell lysates were collected as described according to the RaPID procedure (i.e., beads and unbroken cells were sedimented at 4°C by centrifugation at 1700 x*g* for 1 min, and the supernatant removed to new microcentrifuge tubes and centrifuged at 11,000 x*g* at 4°C for 10 min). After centrifugation, a thin translucent layer of chromatin was observed on the top of the pellet of cell debris. The supernatant was discarded and the pellet resuspended in 1 ml FA lysis buffer. The resuspended material was centrifuged at 13,000 x*g* for 10 min at 4°C and the resulting pellet resuspended in 500 µl of 1.2x *TaqI* restriction enzyme buffer. Next, 7.5 µl of 20% (w/v) SDS (final concentration of 0.3% SDS) was added and the sample incubated for 1 hr at 37°C with shaking (900 rpm). Then, 50 µl of 20% (v/v) Triton X-100 (final concentration = 1.8%) was added and the sample further incubated for 1 hr at 37°C with shaking (900 rpm). An undigested sample of genomic DNA (50 µl aliquot) was removed and stored at −20°C to determine enzyme digestion efficiency. To the remaining sample, 200 U of *TaqI* (New England Biolabs) was added and incubated at 60°C overnight. The next day, 150 µl of digested sample was heat-inactivated at 80°C for 20 min in the presence of added SDS (24 µl 10% SDS; final concentration = 1.7%). To samples of digested material (174 ul each), 626 ul of 1.15x ligation buffer and 80 ul of 10% Triton X-100 (final concentration of 1%) were added and incubated at 37°C for 1 hr, while shaking gently. Proximity ligation in the sample was performed using 100 U of added T4 DNA ligase (New England Biolabs) at 16°C for 16 hr. The ligated samples were then digested with RNase (final concentration of 11 ng/µl; RNaseA and 28 U/µl RNAseT1; ThermoScientific) at 37°C for 20 min. Proteinase K (final concentration of 56 ng/µl; Sigma Aldrich) digestion was performed at 65°C for 1 hr (note: final concentration of SDS = 0.4%). The 3C DNA template was extracted twice using an equal volume of phenol-chloroform (1:1 TE-saturated phenol-chloroform v/v), followed by extraction with an equal volume of phenol-chloroform-isoamyl alchohol (25:24:1 v/v), followed by extraction with an equal volume of chloroform-isoamyl alcohol (24:1 v/v). The DNA was precipitated in the presence of 2 ul of glycogen (20 mg/ml), sodium acetate (0.3 M final concentration, pH 5.2), and 2.5 volumes of ethanol at RT overnight. The 3C DNA template obtained was stored at −20°C. DNA concentration was determined by absorption spectroscopy at 260 nm. Typically, 125–500 ng of the 3C DNA template was used in the PCR reactions. The PCR product was further eluted and sequenced for confirmation.

## Acknowledgements

The authors are grateful to Tsviya Olender (Weizmann Institute of Science) and Jonas Felix (Weizmann Institute of Science) for bioinformatics help, and Aviv Regev (Broad Institute, Cambridge, MA) for helpful support and for coining the term *transperon*. We also thank Amir Aharoni and Daniel Dovrat (Ben Gurion University, Beersheva), and Susan Gasser (Biozentrum, Basel) for strains and plasmids. This work was supported by grants to JEG from the German-Israel Foundation (GIF; #I-1190–96.13/2012) and Minerva Foundation, Germany, as well as the Astrachan Olga Klein Fund (Weizmann Institute of Science) and the Israel Science Foundation (#578/18). CN and JEG were partially supported by funding from NIH (NHGRI U54HG00306). RRN was supported by a VATAT Fellowship for Postdoctoral Fellows from China and India (Israel Council of Higher Education). DZ is employed at Merck, Darmstadt, Germany. MCAD and HSS study at the Free University, Berlin, Germany. HSS works at the Vitalant Research Institute, San Francisco, CA. JEG holds the Besen-Brender Chair in Microbiology and Parasitology (Weizmann Institute of Science).

## Additional information

### Competing interests

Chad Nusbaum: Chad Nusbaum is affiliated with Cellarity Inc. The author has no financial interests to declare. The other authors declare that no competing interests exist.

### Funding

| Funder | Grant reference number | Author |
|---|---|---|
| German-Israeli Foundation for | I-1190-96.13/2012 | Jeffrey E Gerst |

| | | |
|---|---|---|
| Scientific Research and Development | | |
| Minerva Foundation | 711130 | Jeffrey E Gerst |
| Weizmann Institute of Science | | Jeffrey E Gerst |
| National Institutes of Health | NHGRI U54HG00306 | Chad Nusbaum<br>Jeffrey E Gerst |
| Council of Higher Education | | Rita Gelin-Licht |
| Israel Science Foundation | 578/18 | Jeffrey E Gerst |

The funders had no role in study design, data collection and interpretation, or the decision to submit the work for publication.

## Author contributions
Rohini R Nair, Conceptualization, Data curation, Formal analysis, Validation, Investigation, Visualization, Methodology, Writing - original draft, Writing - review and editing; Dmitry Zabezhinsky, Conceptualization, Data curation, Formal analysis, Investigation, Visualization, Methodology, Writing - review and editing; Rita Gelin-Licht, Data curation, Formal analysis, Investigation; Brian J Haas, Data curation, Software, Formal analysis, Writing - original draft; Michael CA Dyhr, Formal analysis, Investigation; Hannah S Sperber, Data curation, Formal analysis; Chad Nusbaum, Resources, Funding acquisition, Writing - original draft; Jeffrey E Gerst, Conceptualization, Supervision, Funding acquisition, Visualization, Writing - original draft, Project administration, Writing - review and editing

## Author ORCIDs
Jeffrey E Gerst https://orcid.org/0000-0002-8411-6881

## Decision letter and Author response
Decision letter https://doi.org/10.7554/eLife.66050.sa1
Author response https://doi.org/10.7554/eLife.66050.sa2

# Additional files

## Supplementary files
• Supplementary file 1. RNA-sequencing data (Excel table). Different MS2 aptamer-tagged mRNAs (listed on column heads) expressed from their genomic loci or control cells (Cntrl) lacking a tagged mRNA were precipitated from *MATα* yeast (BY4742) by MS2-CP-GFP-SBP after formaldehyde cross-linking in vivo and cell lysis procedures (RaPID; see 'Materials and methods'). RNA-seq was performed, and the results for each gene are listed on the following sheets. Sheet 1 (Counts.matrix) represents the number of reads assigned to each gene by Kallisto. Sheet 2 (edgeR_DE_results) represents all genes identified by edgeR, as being differentially expressed in the corresponding pairwise comparison to the control, requiring a false discovery rate (FDR) ≤0.1. Sheet 3 (DE_expr_sub_cntrl) represents gene length-normalized expression values, log2 transformed followed by subtraction of the control expression value. Sheet 4 (Expr_targets_sub_cntrl) represents the sheet three values restricted to the baits (target mRNAs).

• Supplementary file 2. Yeast strains used in this study. In this table, we include the strain name, genotype, and source of origin.

• Supplementary file 3. Plasmids used in this study. In this table, we include the plasmid name, gene expressed, vector backbone used, copy number of the plasmid, selection marker, and source of origin.

• Transparent reporting form

## Data availability
All data is available within the text, figures, and tables of the manuscript.

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
