## [Decision Letter]

**Acceptance summary:**

This work reveals that yeast regulate cellular processes by packaging mRNAs encoding proteins that function on the same pathway into single ribonucleoprotein particles that can be co-trafficked and presumably co-translated. Interestingly, this form of gene regulation is dependent upon histone H4 function and interchromosomal contacts that allow the genes encoding these mRNAs to be closely apposed during transcription. As opposed to prokaryotic cells that employ polycistronic operons, eukaryotic cells can employ monocistronic messages as RNA operons (transperons) to achieve gene co-expression.

**Decision letter after peer review:**

[Editors’ note: the authors submitted for reconsideration following the decision after peer review. What follows is the decision letter after the first round of review.]

Thank you for submitting your work entitled "Multiplexed mRNA assembly into ribonucleoprotein particles plays an operon-like role in controlling yeast physiology" for consideration by *eLife*. Your article has been reviewed by 3 peer reviewers, one of whom is a member of our Board of Reviewing Editors, and the evaluation has been overseen by a Senior Editor. The reviewers have opted to remain anonymous.

Our decision has been reached after consultation between the reviewers. Based on these discussions and the individual reviews below, we regret to inform you that your work will not be considered further for publication in *eLife*.

All three reviewers agreed that your manuscript contains very interesting information, and a description of functional mRNP packaging could be of great importance to the field. However, all three reviewers raise a number of important technical concerns that prevent publication of your manuscript in its current form. Given the extensive nature of the requested revisions, the decision was to 'reject' the paper, in line with the editorial policy of *eLife*. However, we would like to encourage you to perform the requested control experiments and to resubmit a revised manuscript. Although such a revised manuscript would be treated like a new submission, we would attempt to consult the same editors and reviewers in the review process.

*Reviewer #1:*

In this manuscript, Nair et al., propose that functionally related mRNAs can be packaged and transported in an operon-like manner. The discovery of such functional mRNP packages, termed transperons, could have wide-ranging implications for our understanding of the regulation of gene expression. Unfortunately, there are significant technical problems with the results, and while the manuscript contains a large amount of information, it lacks the required rigor, and key controls are missing. Most importantly, additional experiments are needed to demonstrate the existence of specific transperons.

1. Introduction of aptamer-binding sites can lead to the accumulation of decay intermediate that accumulate in P-body-like structures (see results from the Parker and Weis labs 2017). It is therefore critical that the authors perform extensive FISH experiments with untagged alleles (using probes against the body of the mRNAs). In situs should be performed for the different groups of mRNAs involved in mating, heat shock etc. The experiments need to be carefully quantified, statistically evaluated and all experiments need to include negative controls (i.e. mRNAs that do not co-localize). FISH experiments should be performed in wt and in H4 mutants. Of note, it is important to differentiate between nuclear spots, which likely present sites of transcription, and cytoplasmic spots. Only significant cytoplasmic-localization would lend further support for the transperon hypothesis.

2. The RAPID results lack a statistical evaluation. Not all interactions 'look' convincing, are these results significant?

3. The fact that deletion of DHH1 or PAT1 does not affect mating is not an argument against (1). However, the authors should colocalize Dhh1 and Pat1 in their aptamer-tagged strains. Note that this experiment does not replace the need of the FISH experiments described in (1).

4. The nuclear co-localization experiments with Hhf1 protein and the STE2 mRNA need negative controls. Hhf1 seems to fill the entire nucleus and thus is likely to "interact' with every mRNA that would be tested in such a way.

*Reviewer #2:*

In this manuscript the authors present a detailed set of experiments to describe and mechanistically characterize potential multiplexing of functionally related mRNAs. The focus is mainly the mating pathway mRNAs and the physiology of the mating pathway in yeast. The work supports models where the fate of functionally related mRNAs is coordinated and the authors show that this coordination is first seeded at the site of mRNA transcription which may involve a histone protein. The mechanistic advances supporting colocalized mRNA biogenesis at the site of transcription in the nucleus leading to a coordination of the subsequent cytosolic fate of this set of mRNAs are exciting and particularly noteworthy. –

*Reviewer #3:*

The observations presented are potentially interesting. But many of the data presented do not match, in my opinion, the quality standards that one would expect for such experiments in 2020.

A few examples. Only one RaPID-seq experiment is presented for each bait factor (Figure 1, S1 and Table S1). It is only stated "The experiment was performed twice and gave similar results"…… The standard for this type of experiment is rather to perform the experiments in triplicates and perform statistical analyses with programs such as DeSEQ2. The authors say that the replicate gives "similar results", why not provide them in Table S1 then?

Figure 2 A and B provide the data for the key experiment, which is at the basis of the whole manuscript. One should expect that this would be very robust data. In fact, this is not the case at all. The individual mRNA enrichments are not quantified by qRT-PCR, as one would expect, but by a single ethidium bromide stained agarose gel. Even worse, I am actually unable to see what the authors claim, i.e. a signal for the STE3, SRO7, OM45 and EXO70 PCR products in the IP fractions. The only signals one can see are for MFalpha1 and MFalpha2, but the signals in the input fraction are much more abundant for those genes that the signals seen for the control transcripts, making it impossible to say if these mRNAs are actually truly enriched. The control used for Figure A is the signal from the input, while for B, the only control provided is "-RT". How one would expect to get an idea of an enrichment level without the input data and the proper control mRNAs. Frankly, I don't understand. These experiments should be analyzed by qRT-PCR with the proper replicates and control mRNAs to be able to quantify enrichment values and calculate standard deviations. These remarks stand for Figure 3AandB, Figure 4C etc.

For the Fish experiments, some field view, with several cells, should be provided in the supplemental data.

Figure 6A: "Mating mRNAs and histone H4 co-localize only in the nucleus." I don't understand the meaning of this. Hhf1 localizes to the whole nucleus (see the yeast-GFP localization data base). Hence, in this type of experiment, Hhf1 will merely localize the nucleus (pretty much as dapy staining). How can one say that an mRNA fish signal, appearing as a dot, "co-localize" with Hhf1. It only says that the nuclear fraction of this mRNA is…… nuclear!

I will stop here. I think it is pretty clear that the authors should repeat properly quite a number of experiments before one would consider publishing this work.

[Editors’ note: further revisions were suggested prior to acceptance, as described below.]

Thank you for submitting your article "Multiplexed mRNA assembly into ribonucleoprotein particles plays an operon-like role in yeast cell physiology" for consideration by *eLife*. Your article has been reviewed by 3 peer reviewers, one of whom is a member of our Board of Reviewing Editors, and the evaluation has been overseen by Kevin Struhl as the Senior Editor. The reviewers have opted to remain anonymous.

Essential Revisions:

The reviewers are concerned about the reproducibility of some of the experiments that are shown in the paper. The concerns are described in their detailed comments below but key issues include:

1. Figure 2

Given the high variability the results remain difficult to interpret.

2. AID experiments

The AID experiments are problematic as the mRNA levels go down and tagging alone in the absence of auxin leads to a protein reduction.

3. Colocalization microscopy data

The mRNA localization results vary dramatically between different experiments, which makes it difficult to interpret these results.

*Reviewer #1:*

In this revised paper, the authors have added additional controls and statistical analyses. This has improved the quality of the manuscript. However, there are still significant technical issues.

1. Figure 2A/B

Based on the figure legend it remains unclear what the authors call significant versus not significant?

2. Figure 2, 3, 8, 9

The mRNA localization results vary dramatically between different experiments (e.g., STE2 foci in different conditions). Why? Irrespectively, this makes it very hard to compare the experiments and the controls. Furthermore, the signal overlap in hs between SSA2 and HSP104 are not convincing. It could be a consequence of the enhance HSP104 expression.

3. The AID experiments in hs are not convincing. mRNA levels go down which might be sufficient to reduce co-localization (see 2). AID-tagging alone (without auxin) also leads to a reduction, which is potentially problematic.

4. Figure 9 (A-C) is also problematic since no convincing negative controls are included.

*Reviewer #2:*

In this manuscript the authors present a detailed set of experiments to describe and mechanistically characterize potential multiplexing of functionally related mRNAs. The focus is mainly the mating pathway mRNAs and the physiology of the mating pathway in yeast. The work supports models where the fate of functionally related mRNAs is coordinated and the authors show that this coordination is first seeded at the site of mRNA transcription which may involve a histone protein. The mechanistic advances supporting colocalized mRNA biogenesis at the site of transcription in the nucleus leading to a coordination of the subsequent cytosolic fate of this set of mRNAs are exciting and particularly noteworthy.

The authors present a revised version of a manuscript describing mRNA multiplexing and going on to study the mechanism of mRNA localization and physiological role of this process. The manuscript is very much improved over the previous version, but I do have some further comments for the authors – detailed below.

1. I wasn't convinced that the IST2 mRNA was significantly enriched in the pulldown in

Figure 2A relative to Figure 2B- as claimed by the authors. Plus many of the error bars On Figure 2B are very high across this analysis- so it is difficult to interpret.

2. Are the images in Figure 7C from a single focal plane? I'm guessing so since some of the cells don't have a spot in their nucleus. If this is the case the authors should add this information to the legend to prevent confusion.

3. Have the authors tested the expression levels of the Hhf1K-R and K-Q- as these mutations a differential effect on protein stability could explain the observations?

4. Similar to above- have the authors measured the RNA levels for AGA2 47-MUT or AGA2 27-MUT relative to wild type, as effects on RNA stability or expression could explain the observed effects.

*Reviewer #3:*

In this novel version of the manuscript, the authors have, I think, satisfactorily answered to most reviewer concerns about the first version.

Considering the suggestion to provide the results of the replicate of the Rapid-RNAseq experiment, the authors rather preferred to remove all mentions of it. Considering that all analyzed and discussed results of the now single documented experiment are validated by independent RaPID followed by qRT-PCR experiments, one can consider this as acceptable.

I have one suggestion, thought. When examining Figure 1 and the associated Table S1, one can see that some of the transcripts most highly enriched together with the bait MS2 tagged mRNAs are actually the mRNAs from the directly adjacent genes. This is true for PET122 (YER153C) pulled down by OXA1 (YER154W), or SPE1 (YKL184W) / ASH1 (YKL185W), SNC1 (YAL030W) / MYO4 (YAL029C), SNC2 (YOR327C) / MYO2 (YOR326W) or ALY2 (YJL084C) / EXO70 (YJL085W). This striking observation, which cannot have escaped the authors, is never mentioned nor discussed, which, I think, is a pity. These genes are most likely without any functional link with the bait mRNAs. Moreover, they are indifferently in tandem, divergent of convergent orientations, which preclude the fact that these co-purifications might result from the existence of some kind of polycistronic mRNAs. The most likely explanation that comes to mind is thus that the RaPID procedure is able to enrich not only MRNP particles but also chromatin fragments with their associated nascent transcripts.

I think that this is important to discuss because it suggests that a substantial fraction of the transcripts enriched in the genomic experiment might actually represent such chromatin-associated transcripts. This observation would not be in conflict with the suggestion in the following sections of the manuscript that co-purified mRNAs are synthesized within "transcription factories" in which the genomic locations of the corresponding scattered genes are brought to vicinity.

Anyhow, this observation, which is readily apparent, should be discussed.

---

## [Author Response]

[Editors’ note: the authors resubmitted a revised version of the paper for consideration. What follows is the authors’ response to the first round of review.]

Reviewer #1:In this manuscript, Nair et al.et al., propose that functionally related mRNAs can be packaged and transported in an operon-like manner. The discovery of such functional mRNP packages, termed transperons, could have wide-ranging implications for our understanding of the regulation of gene expression. Unfortunately, there are significant technical problems with the results, and while the manuscript contains a large amount of information, it lacks the required rigor, and key controls are missing. Most importantly, additional experiments are needed to demonstrate the existence of specific transperons.

We thank the reviewer for their comments and have revised the paper accordingly with new added experiments that validate our previous observations.

1. Introduction of aptamer-binding sites can lead to the accumulation of decay intermediate that accumulate in P-body-like structures (see results from the Parker and Weis labs 2017). It is therefore critical that the authors perform extensive FISH experiments with untagged alleles (using probes against the body of the mRNAs). In situs should be performed for the different groups of mRNAs involved in mating, heat shock etc. The experiments need to be carefully quantified, statistically evaluated and all experiments need to include negative controls (i.e. mRNAs that do not co-localize). FISH experiments should be performed in wt and in H4 mutants. Of note, it is important to differentiate between nuclear spots, which likely present sites of transcription, and cytoplasmic spots. Only significant cytoplasmic-localization would lend further support for the transperon hypothesis.

We appreciate the reviewer’s comment and, per their suggestion, performed smFISH experiments (with statistical analyses) on strains expressing the untagged endogenous alleles in both WT and in H4 deletion mutants, and using specific probes against the mRNAs (Figures 3A, 8B, S3E, and S8C) along with different negative controls. The results fully support our previous observations made using the endogenously expressed aptamer-tagged mRNAs. RNA colocalization was observed both in the nucleus and cytoplasm, further supporting our hypothesis of the existence of these RNA multiplexes/transperons.

2. The RAPID results lack a statistical evaluation. Not all interactions 'look' convincing, are these results significant?

Since this is a critical observation of our paper, and taking into consideration the reviewer’s comment, we re-performed the RaPID pulldown experiments (shown previously in the original Figure 2A and B) using both MATa and MATalpha yeast strains and now used qRT-PCR to obtain quantitative results regarding the pulldown of the mating pathway mRNAs. These are shown in the new Figure 2A and B. We found that the results matched our original RT-PCR results and show that the quantitative coprecipitation of mating RNAs is significant compared to the negative controls (Figure 2, and B). We note that we previously used qRT-PCR to demonstrate the multiplexing of the HSP, MOMP, MAPK RNAs (Figures 8A, 9A and D, S8A and B) in the original version of the manuscript.

3. The fact that deletion of DHH1 or PAT1 does not affect mating is not an argument against (1). However, the authors should colocalize Dhh1 and Pat1 in their aptamer-tagged strains. Note that this experiment does not replace the need of the FISH experiments described in (1).

As per the reviewer’s suggestion we looked for the colocalization of Dhh1 protein and MS2-tagged STE2 mRNA under conditions of normal growth (and mating), as compared to conditions that induce P-body formation (i.e. no glucose – 10min; Figure S5C). Under normal growth conditions (2% glucose), we found that only ~2% of cells had Dhh1-mCherry-labeled puncta and which did not colocalize with the mating pathway mRNA examined (STE2). As a positive control for P-body formation, we exposed the cells to no glucose conditions for 10min, which resulted in all cells having well-formed Dhh1-mCherry puncta (P-bodies). Under these conditions, we now observed around 41% colocalization of puncta colocalizing with STE2 mRNA (Figure S5C). Our results confirm the idea that under conditions conducive to normal growth or mating (i.e. in the presence of glucose) that mating pathway mRNAs do not colocalize with P-bodies (if even formed) and, therefore, conclude that P-bodies cannot be a major site for mating pathway signaling events. Thus, our work contrasts with the earlier study of Aronov et al.et al. 2015 (JCB) in a number of critical points.

4. The nuclear co-localization experiments with Hhf1 protein and the STE2 mRNA need negative controls. Hhf1 seems to fill the entire nucleus and thus is likely to "interact' with every mRNA that would be tested in such a way.

We concur with the reviewer in that an mRNA in the nucleus is likely to colocalize with Hhf1 protein because of the high concentration of protein therein. The point we endeavored to show is that we cannot observe colocalization between RFP-Hhf1 with the mRNAs in the cytoplasmic transperons, either due to lack of RFP-Hhf1 presence in the cytoplasm or that the level in transperons is below the threshold of detection. Since the experiment is inconclusive, we have removed this figure in order to prevent any confusion.

Reviewer #3:The observations presented are potentially interesting. But many of the data presented do not match, in my opinion, the quality standards that one would expect for such experiments in 2020.

We thank the reviewer for their comments and have revised the paper with new experiments that validate our previous observations.

A few examples. Only one RaPID-seq experiment is presented for each bait factor (Figure 1, S1 and Table S1). It is only stated "The experiment was performed twice and gave similar results"…… The standard for this type of experiment is rather to perform the experiments in triplicates and perform statistical analyses with programs such as DeSEQ2. The authors say that the replicate gives "similar results", why not provide them in Table S1 then?

Since the RaPID-seq experiments performed were similar, but non-identical (i.e. in that they did not use all the same baits), we did not put them in the table nor perform statistical comparisons. We have now removed this sentence (i.e. about the experiment being performed twice) from the text in order to prevent any confusion. We concur with the reviewer that if these experiments would be performed today they would be done differently, with replicates, and under different conditions of stringency in order to identify (and not eliminate) other potential multiplexes. We are in the process of setting this up and, in parallel, will perform quality Hi-C experiments for the yeast genome under different conditions, since allelic coupling appears to be part of the mechanism leading to RNA multiplexing. In this revised version of the paper we now validate the existence of mating pathway RNA multiplexes (as observed in Figure 1A and B for MATalpha cells), in both MATa and MATalpha yeast using RaPID and qRT-PCR (Figure 2A and B). These results verify the existence of the multiplex shown in Figure 1 as well as the original RT-PCR results obtained via RaPID pulldowns in the original Figure 2. In addition, we now add smFISH experiments (new Figure 3A and B) that also validate our original finding that aptamer-tagged mating pathway mRNAs colocalize (Figure 2D-F). Together with the original data, our new experimentation further validates the hypothesis that these mRNAs form functional multiplexes in yeast.

Figure 2 A and B provide the data for the key experiment, which is at the basis of the whole manuscript. One should expect that this would be very robust data. In fact, this is not the case at all. The individual mRNA enrichments are not quantified by qRT-PCR, as one would expect, but by a single ethidium bromide stained agarose gel. Even worse, I am actually unable to see what the authors claim, i.e. a signal for the STE3, SRO7, OM45 and EXO70 PCR products in the IP fractions. The only signals one can see are for MFalpha1 and MFalpha2, but the signals in the input fraction are much more abundant for those genes that the signals seen for the control transcripts, making it impossible to say if these mRNAs are actually truly enriched. The control used for Figure A is the signal from the input, while for B, the only control provided is "-RT". How one would expect to get an idea of an enrichment level without the input data and the proper control mRNAs. Frankly, I don't understand. These experiments should be analyzed by qRT-PCR with the proper replicates and control mRNAs to be able to quantify enrichment values and calculate standard deviations. These remarks stand for Figure 3AandB, Figure 4C etc.

We concur with the reviewer, and as multiplexing is a critical observation, we reperformed the RaPID pulldowns of select target mRNAs from both MATa and MATalpha cells and quantitated the results by qRT-PCR (versus a wild-type strain expressing the coat protein alone). The results verify our previous RT-PCR experiments (now removed) showing quantitative and statistically significant results regarding the specific coprecipitation of mating RNAs, as compared to negative controls (new Figure 2A and B).

For the Fish experiments, some field view, with several cells, should be provided in the supplemental data.

We have included Author response image 1 and Author response image 2 field views of some experiments, as requested. We did not include these in the manuscript, so as not to further increase its length. As can be seen, we observed similar results in all cells scored and the representative cells presented in the figures are accurate.

**Author response image 1. respfig1:** HS – heat shocked cells; NHS – non-heat shocked cells. Note colocalization of HSP104 mRNA with SSA2 mRNA, but not with STE2 mRNA.

**Author response image 2. respfig2:** note the colocalization between AGA1 mRNA and STE2 mRNA, but not with HSP150 mRNA.

Figure 6A: "Mating mRNAs and histone H4 co-localize only in the nucleus." I don't understand the meaning of this. Hhf1 localizes to the whole nucleus (see the yeast-GFP localization data base). Hence, in this type of experiment, Hhf1 will merely localize the nucleus (pretty much as dapy staining). How can one say that an mRNA fish signal, appearing as a dot, "co-localize" with Hhf1. It only says that the nuclear fraction of this mRNA is…… nuclear!

We concur with the reviewer’s comment. We endeavored to show whether Hhf1 could colocalize with a mating mRNA in the cytoplasm, but were unable to do so, perhaps, due to the low intensity of the signal in the cytoplasm. Given that the experiment was inconclusive in this regard we have now removed it from the manuscript. Obviously, colocalization in the nucleus was pretty much a given.

I will stop here. I think it is pretty clear that the authors should repeat properly quite a number of experiments before one would consider publishing this work.

We thank the reviewer for the critique and have taken their comments (as well as those of the other reviewers) to heart, and have reperformed a number of experiments using different techniques (e.g. qRT-PCR, smFISH). Importantly, using all manners of approach we could validate our previous findings that show the formation of these mRNA complexes, their presence in the cytoplasm, and their importance to cell physiology.

[Editors’ note: what follows is the authors’ response to the second round of review.]

Essential Revisions:The reviewers are concerned about the reproducibility of some of the experiments that are shown in the paper. The concerns are described in their detailed comments below but key issues include:1. Figure 2Given the high variability the results remain difficult to interpret.

As one of the original pulldown experiments showed variability, we have repeated this RaPID experiment and added the data to a new graphic presented in the Figure 2A and B. This new figure shows improved significance in the pulldown results for the mating pathway mRNAs and supports the idea that each yeast haplotype creates its own mating RNA multiplex.

2. AID experimentsThe AID experiments are problematic as the mRNA levels go down and tagging alone in the absence of auxin leads to a protein reduction.

We apologize for any confusion regarding the AID experiments. As shown previously in Figure S3D (now referred as *Figure 4—figure supplement 1D)*, the mRNA levels for both *STE2* and *AGA1* do not go down upon auxin addition (either with or without α factor). In fact, the levels for *STE2* mRNA do not change, while those of *AGA1* actually go up a bit. Thus, the depletion in histone H4 does not greatly affect the amounts of the mating pathway mRNAs. Therefore, the reduced colocalization of *STE2* and *AGA1* mRNA (Figure S3E; now referred as *Figure 4—figure supplement 1E*) is independent of RNA levels.

In new *Figure 8—figure supplement 2D*, we now show that upon histone depletion the mRNA levels of *HSP104* and *SSA2* do not go down, either with or without heat shock. This shows that the reduced amount of *HSP104* and *SSA2* mRNA colocalization (S8C; now referred as *Figure 8—figure supplement 2C*) is also independent of RNA levels. We also note that AID tagging alone does not lead to a reduction in histone H4 (Hhf1), as shown previously in Figure S3B; now referred to as *Figure 4—figure supplement 4B*.

3. Colocalization microscopy dataThe mRNA localization results vary dramatically between different experiments, which makes it difficult to interpret these results.

We concur that some variability in the number of fluorescent labeled spots/foci was observed in the different smFISH experiments. However, it is important to note these results did not change our overall conclusions regarding RNA colocalization. Nevertheless, we have now provided a new smFISH experiment (new Figure 2E and F) for *STE2* and *AGA2* mRNA using probes to native *STE2* and PP7 aptamer-based probes for *AGA2*, which show greatly increased numbers of both RNA spots (foci) and colocalized RNAs in the nucleus and in the cytoplasm. This replaces the original and much older Figure 2E and F, which differed in spot number from the other smFISH

Reviewer #1:In this revised paper, the authors have added additional controls and statistical analyses. This has improved the quality of the manuscript. However, there are still significant technical issues.

We thank the reviewer for their helpful comments and suggestions, and have addressed the issues raised.

1. Figure 2A/BBased on the figure legend it remains unclear what the authors call significant versus not significant?

We concur with the reviewer’s comment and now better illustrate the significance, statistical tests done, and number of experiments performed in the figure and legend of Figure 2A and B. We apologize for their omission in the previous version of the manuscript. In addition, we have added new data that shows increased statistical significance of the mating RNAs pulled down by the target RNA.

2. Figure 2, 3, 8, 9The mRNA localization results vary dramatically between different experiments (e.g., STE2 foci in different conditions). Why? Irrespectively, this makes it very hard to compare the experiments and the controls. Furthermore, the signal overlap in hs between SSA2 and HSP104 are not convincing. It could be a consequence of the enhance HSP104 expression.

We concur with the reviewer’s comment regarding variability (in the number of fluorescent-labeled foci) used to score colocalization. However, it is important to note these results did not change our overall conclusions regarding RNA colocalization. Nevertheless, we repeated the smFISH experiment in Figure 2E and F now using probes for endogenous *STE2* mRNA and PP7 probes for *AGA2* mRNA, and observed much greater numbers of RNA foci and colocalization. The inclusion of this new data greatly reduces the variability between Figure 2E and F and Figures 3A and B and 8B and C.

Regarding the greater level of RNA colocalization between *SSA2* and *HSP104* upon heat shock (Figure 8B and C), we concur that this could result due to enhanced gene expression, as we now show to occur Figure 8—figure supplement 2D. However, we also showed previously (Figure 8A) that heat shock increases *SSA2* and *HSP104* multiplexing. Moreover, we show in Figure 8—figure supplement 2C that colocalization is dependent on histone H4 levels, which is independent of mRNA levels as shown in Figure 8—figure supplement 2D. Therefore, the increased colocalization is most likely a consequence of increased multiplexing and not increased gene expression.

3. The AID experiments in hs are not convincing. mRNA levels go down which might be sufficient to reduce co-localization (see 2). AID-tagging alone (without auxin) also leads to a reduction, which is potentially problematic.

As shown previously in Figure S3D (now referred as *Figure 4—figure supplement 1D)*, the mRNA levels for both *STE2* and *AGA1* do not go down upon auxin addition (either with or without α factor). In fact, the levels for *STE2* mRNA do not change, while those of *AGA1* actually go up a bit. Thus, the auxin-induced depletion of histone H4 does not greatly affect the levels of the mating pathway mRNAs and, therefore, the reduced colocalization of *STE2* and *AGA1* mRNAs (Figure S3E; now referred as *Figure 4—figure supplement 1E*) is independent of RNA levels. In *Figure 8—figure supplement 2D*, we now show that the mRNA levels of *HSP104* and *SSA2* also do not go down upon histone depletion, either with or without heat shock. This shows that the reduced amounts of *HSP104* and *SSA2* mRNA colocalization (S8C; now referred as *Figure 8—figure supplement 2C*) are also likely to be independent of RNA levels.

As shown in *Figure 8—figure supplement 2D*, AID tagging alone (e.g. without auxin) does not affect the levels of *HSP104* and *SSA2* mRNA. Likewise AID tagging (e.g. with α factor) did not affect *STE2* mRNA levels, but slightly increased those of *AGA1* slightly (Figure S3D; now referred as *Figure 4—figure supplement 1D*). Thus, AID tagging did not affect gene expression in the case of either transperon. We also showed previously that AID tagging alone did not affect Hhf1 protein levels (Figure S3B; now referred as *Figure 4—figure supplement 1B*). We did observe a decrease in RNA colocalization in the tagged strains, but this more likely reflects the effect of the single histone H4 deletion (*hhf2*delta), which we have demonstrated to affect RNA multiplexing (Figure 4C and D).

4. Figure 9 (A-C) is also problematic since no convincing negative controls are included.

As per the reviewer’s suggestion we have now included negative controls (*e.g. HSP104* vs. *TOM6* and *HSP104* vs. *FUS3* RNA colocalization) for the mitochondrial outer membrane and MAPK pathway transperons, respectively. Our results fully support our previous observations regarding the existence of these RNA multiplexes/transperons.

Reviewer #2:In this manuscript the authors present a detailed set of experiments to describe and mechanistically characterize potential multiplexing of functionally related mRNAs. The focus is mainly the mating pathway mRNAs and the physiology of the mating pathway in yeast. The work supports models where the fate of functionally related mRNAs is coordinated and the authors show that this coordination is first seeded at the site of mRNA transcription which may involve a histone protein. The mechanistic advances supporting colocalized mRNA biogenesis at the site of transcription in the nucleus leading to a coordination of the subsequent cytosolic fate of this set of mRNAs are exciting and particularly noteworthy.The authors present a revised version of a manuscript describing mRNA multiplexing and going on to study the mechanism of mRNA localization and physiological role of this process. The manuscript is very much improved over the previous version, but I do have some further comments for the authors – detailed below.

We thank the reviewer for their helpful comments and suggestions, and have addressed the issues raised.

1. I wasn't convinced that the IST2 mRNA was significantly enriched in the pulldown inFigure 2A relative to Figure 2B- as claimed by the authors. Plus many of the error bars On Figure 2B are very high across this analysis- so it is difficult to interpret.

As one of the original pulldown experiments showed variability, we repeated this RaPID experiment and have added the data to a new graphic presented in the Figure 2A and B. This new figure shows improved significance in the pulldown results for the mating pathway mRNAs and supports the idea that each yeast haplotype creates its own mating RNA multiplex. We now better describe the significance, statistical tests performed, and number of experiments performed in the figure and legend of Figure 2A and B. With regards to *IST2*, we could only observe significant enrichment in the *SAG1* pulldown from *MAT*alpha cells. Since we have not done any other experiments with *IST2* this result is inconclusive at this point.

2. Are the images in Figure 7C from a single focal plane? I'm guessing so since some of the cells don't have a spot in their nucleus. If this is the case the authors should add this information to the legend to prevent confusion.

For all microscopy figures a single focal plane is shown. We concur with the reviewer and now mention this in the various figure legends.

3. Have the authors tested the expression levels of the Hhf1K-R and K-Q- as these mutations a differential effect on protein stability could explain the observations?

We concur with reviewer’s suggestion and performed qRT-PCR to examine the levels of the endogenously expressed *HHF1*1K-R mutant used in Figure 5C. We observed no statistical difference in *HHF1* mRNA levels between the WT and *HHF1*K-R mutant strains, and now include mention of this observation in the Results section. We did not examine protein levels (due to a lack of specific antibody), so it’s possible that protein instability could also contribute to the effect. A statement to this effect has been added to the text. As for the *HHF1*K-Q mutation (used in Figure 5B), this gene was expressed using a multicopy plasmid, but we did not examine protein stability.

4. Similar to above- have the authors measured the RNA levels for AGA2 47-MUT or AGA2 27-MUT relative to wild type, as effects on RNA stability or expression could explain the observed effects.

We performed qRT-PCR to measure *AGA2* 47-MUT or *AGA2* 27-MUT RNA levels in mutant backgrounds (used in Figure 6B and C) and observed no statistical difference between WT *AGA2* and the *AGA2* 47-MUT mutant strain, but did see an increased expression of *AGA2* 27-Mut. This may explain why this mutant had a lesser effect on multiplexing and mating. A note regarding this observation has now been added to the Results section.

Reviewer #3:In this novel version of the manuscript, the authors have, I think, satisfactorily answered to most reviewer concerns about the first version.

We thank the reviewer for their helpful comment and have addressed the issue raised.

Considering the suggestion to provide the results of the replicate of the Rapid-RNAseq experiment, the authors rather preferred to remove all mentions of it. Considering that all analyzed and discussed results of the now single documented experiment are validated by independent RaPID followed by qRT-PCR experiments, one can consider this as acceptable.I have one suggestion, thought. When examining Figure 1 and the associated Table S1, one can see that some of the transcripts most highly enriched together with the bait MS2 tagged mRNAs are actually the mRNAs from the directly adjacent genes. This is true for PET122 (YER153C) pulled down by OXA1 (YER154W), or SPE1 (YKL184W) / ASH1 (YKL185W), SNC1 (YAL030W) / MYO4 (YAL029C), SNC2 (YOR327C) / MYO2 (YOR326W) or ALY2 (YJL084C) / EXO70 (YJL085W). This striking observation, which cannot have escaped the authors, is never mentioned nor discussed, which, I think, is a pity. These genes are most likely without any functional link with the bait mRNAs. Moreover, they are indifferently in tandem, divergent of convergent orientations, which preclude the fact that these co-purifications might result from the existence of some kind of polycistronic mRNAs. The most likely explanation that comes to mind is thus that the RaPID procedure is able to enrich not only MRNP particles but also chromatin fragments with their associated nascent transcripts.I think that this is important to discuss because it suggests that a substantial fraction of the transcripts enriched in the genomic experiment might actually represent such chromatin-associated transcripts. This observation would not be in conflict with the suggestion in the following sections of the manuscript that co-purified mRNAs are synthesized within "transcription factories" in which the genomic locations of the corresponding scattered genes are brought to vicinity.Anyhow, this observation, which is readily apparent, should be discussed.

We duly appreciated the reviewer’s suggestion. This was indeed observed, but since we had not examined the interactions could not prove or disprove the findings. However, we agree with the reviewer’s suggestion and explanation, and briefly discuss the issue in the Discussion section.